# Ensuring User-side Fairness in Dynamic Recommender Systems

## ABSTRACT

User-side group fairness is crucial for modern recommender systems, aiming to alleviate performance disparities among user groups defined by sensitive attributes like gender, race, or age. In the ever-evolving landscape of user-item interactions, continual adaptation to newly collected data is crucial for recommender systems to stay aligned with the latest user preferences. However, we observe that such continual adaptation often exacerbates performance disparities. This necessitates a thorough investigation into user-side fairness in dynamic recommender systems, an area that has been unexplored in the literature. This problem is challenging due to distribution shift, frequent model updates, and non-differentiability of ranking metrics. To our knowledge, this paper presents the first principled study on ensuring user-side fairness in dynamic recommender systems. We start with theoretical analyses on fine-tuning v.s. retraining, showing that the best practice is incremental fine-tuning with restart. Guided by our theoretical analyses, we propose FAir Dynamic rEcommender (FADE), an end-to-end fine-tuning framework to dynamically ensuring user-side fairness over time. To overcome the non-differentiability of recommendation metrics in the fairness loss, we further introduce Differentiable Hit (DH) as an improvement over the recent NeuralNDCG method, not only alleviating its gradient vanishing issue but also achieving higher efficiency. Besides that, we also address the instability issue of the fairness loss by leveraging the competing nature between the recommendation loss and the fairness loss. Through extensive experiments on real-world datasets, we demonstrate that FADE effectively and efficiently reduces performance disparities with little sacrifice in the overall recommendation performance.

## CCS CONCEPTS

• **Information systems** → **Data mining**; • **Computing methodologies** → *Machine learning*.

## KEYWORDS

recommender systems, user-side fairness, dynamic updates

**ACM Reference Format:**

Anonymous Author(s). 2024. Ensuring User-side Fairness in Dynamic Recommender Systems. In *Proceedings of the ACM Web Conference 2024 (WWW '24), May 13–17, 2024, Singapore.* ACM, New York, NY, USA, 19 pages. https://doi.org/XXXXXXX.XXXXXXX

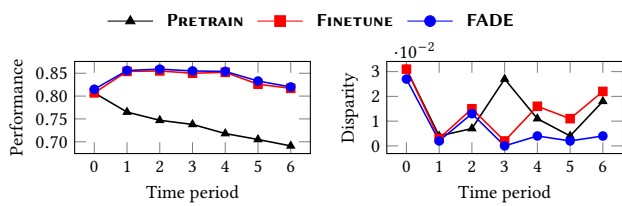

**(a)** Recommendation performance over time    **(b)** Performance disparity between user groups

**Figure 1: Even though incremental fine-tuning with new data (red curve) upholds recommendation performance compared to pretrain (black curve), the disparity gradually expands over time without fairness regularization. (See §4 for detail.)**

## 1 INTRODUCTION

Recommender systems are essential for delivering high-quality personalized recommendations in a two-sided market (i.e., user-side and item-side) [36, 42]. In this market, users provide feedback on recommended items, and the system refines the recommendations to better reflect their preferences. However, these recommender systems can perform poorly for users from certain demographic groups even while delivering high-quality recommendations on average [4, 33]. For example, a job recommender system might recommend more irrelevant job opportunities to female engineers in STEM (Science, Technology, Engineering, and Mathematics), which can significantly impact their career growth [11, 16]. Thus, it is important to alleviate the performance disparity between different user groups in recommender systems [17].

Although there is a parallel line of research on item-side fairness, those methods do not apply to user-side fairness due to the fundamental distinction between user- and item-side fairness. In essence, user-side fairness is concerned with ensuring equitable recommendation quality for different users, while item-side fairness focuses on providing equal exposure opportunities for items within recommendations, often addressing the so-called popularity bias of items through debiasing techniques. For example, several works for item-side fairness [30, 31, 40, 41] calibrates predicted ratings with item popularity, which does not apply to user-side fairness.

Furthermore, due to the evolving nature of user-item interactions, real-world recommender systems continually adapt to new data over time to improve recommendation quality [14, 38]. However, as shown in Fig. 1, neglecting fairness during dynamic adaptation leads to performance disparity between user groups persisting or even expanding over time. This highlights the importance of maintaining user-side fairness in dynamic recommendation.

Despite its critical importance, to the best of our knowledge, user-side fairness [7, 17] has not been explored in the context of dynamic recommendation, which is in stark contrast to the extensive research effort on item-side fairness in dynamic recommendation [8, 23, 40]. As item-side methods are inapplicable to user-side fairness, a thorough study of user-side fairness in dynamic recommendation will substantially expand the frontiers of fair dynamic recommendation and establish a prospective foundation for future research on two-sided fairness [5, 37] in dynamic recommendation.

This paper presents the first principled study of user-side fairness in dynamic recommender systems. We identify and address the following challenges: **(C1) Distribution shift.** Constant emergence of new users/items and evolving user preferences lead to distribution shift. Distribution shift not only affects recommendation performance but also worsens performance disparity among user groups over time. **(C2) Frequent model updates.** Due to distribution shift in dynamic recommendation, recommender systems need frequent updates to cater to current user preferences. This imposes efficiency requirements on the model updating method. However, existing postprocessing methods involve time-intensive re-ranking [7, 17], which are inefficient for frequent model updates. **(C3) Non-differentiability of ranking metrics.** The sorting operation in ranking metrics is non-differentiable. This raises a critical challenge in end-to-end training because we cannot directly use the non-differentiable performance disparity as the fairness loss. Even if one resorts to postprocessing methods like re-ranking [7, 17] which does not involve end-to-end training, they critically suffer from the existing performance disparity in candidate item lists.

To address the challenges, we propose FAir Dynamic rEcommender (FADE), an end-to-end framework employing an incremental fine-tuning strategy to dynamically alleviate performance disparity between user groups. Specifically, our key contributions are:

- **Problem.** We observe that the user-side performance disparity tends to persist or worsen over time, despite improvements in recommendation performance. To our knowledge, we are the first to study user-side fairness in dynamic recommendation.
- **Theory.** To ground the design of our method, we theoretically analyze *fine-tuning* v.s. *retraining* in terms of generalization error (recommendation & fairness) under distribution shift. Our Theorems 3.1 & 3.2 show that the best practice is incremental fine-tuning with restart.
- **Algorithm.** Based on theoretical analyses, we propose FADE, a novel dynamic recommender based on incremental fine-tuning that balances both recommendation quality and user-side fairness. To overcome the non-differentiability of recommendation metrics in the fairness loss, we further introduce *Differentiable Hit* (DH) as an improvement over the recent NeuralNDCG method [25], not only alleviating its gradient vanishing but also achieving higher efficiency. Besides that, we also address the instability of the fairness loss by leveraging the competing nature between the recommendation loss and the fairness loss (Proposition 3.3).
- **Experiments.** Empirical experiments on real-world datasets demonstrate that FADE effectively reduces performance disparity (with an average decrease of 48.91%) without significantly compromising overall performance over time (with an average drop of 2.44%). It also operates efficiently, achieving an 11x faster running time on average compared to the retraining approach.

## 2 PROBLEM DEFINITION

In this section, we first present the key notations in the paper. Then we provide preliminaries on the settings of dynamic recommendation and user-side fairness. Finally, we formally define the problem of dynamic user-side fairness in recommender systems.

**Notations.** Table 1 provides a list of our symbols. Throughout the paper, we use bold upper-case letters for matrices (e.g., $\mathbf{Y}$), bold lower-case letters for vectors (e.g., $\mathbf{r}$) and calligraphic letters for sets

**Table 1: Main symbols used in this paper.**

| Symbol | Description |
|---|---|
| $\mathcal{D}_t$ | Dataset collected at time period $t$ |
| $\mathcal{U}_t, \mathcal{I}_t, \mathcal{E}_t$ | Sets of users, items, and their interactions at time period $t$ |
| $\mathbf{Y}_t$ | User-item interaction matrix at the time period $t$ |
| $\widehat{\mathbf{Y}}_t$ | User-item predicted score matrix at time period $t$ |
| $\mathcal{W}_t$ | Set of model parameters at time period $t$ |
| $a$ | Binary sensitive attribute of a user |
| $\mathcal{L}_{\text{rec}}, \mathcal{L}_{\text{fair}}$ | Recommendation loss and fairness loss, respectively |
| $C_u, N$ | Set of candidate items for a user $u$ and its size |
| $\mathbf{s}_u$ | Unsorted list of recommendation scores of items in $C_u$ |
| $\mathbf{r}_u$ | List of items in $C_u$ ranked by their scores in $\mathbf{s}_u$ |
| $\mathbf{P}_u, \widehat{\mathbf{P}}_u$ | Permutation matrix and relaxed permutation matrix for $\mathbf{s}_u$ |
| $\lambda$ | Scaling parameter for $\mathcal{L}_{\text{fair}}$ |
| $\tau$ | Temperature parameter for $\widehat{\mathbf{P}}_{\mathbf{s}_u}$ |
| $\mu$ | The number of negative items in $C_u$ |
| $n$ | The number of negative items for $\mathcal{L}_{\text{rec}}$ |

(e.g., $\mathcal{U}$). We use standard conventions for indexing. For example, $\mathbf{Y}[i, j]$ is the entry at the $i$-th row and the $j$-th column in matrix $\mathbf{Y}$.

We use $\mathcal{D}_t = \{\mathcal{U}_t, \mathcal{I}_t, \mathcal{E}_t, \mathbf{Y}_t\}$ to denote the dataset collected at time period $t$ $\forall t \in \{1, \ldots, T\}$, [1] where the subscript $_t$ indicates the time period $t$, $\mathcal{U}_t$ is the user set, $\mathcal{I}_t$ is the item set, $\mathcal{E}_t$ is the user-item interaction set, and $\mathbf{Y}_t$ is the user-item interaction matrix. We consider binary user-item interaction in this work, i.e., $\mathbf{Y}_t[u, i] = 1$ if user $u$ has interacted with item $i$ within the $t$-th time period, and 0 otherwise. The initial user set, item set, user-item interaction set, and the user-item interaction matrix before the first time period (i.e., $t = 1$) is denoted as $\mathcal{U}_0, \mathcal{I}_0, \mathcal{E}_0$, and $\mathbf{Y}_0$, respectively. Lastly, we denote the subscript $_{:t}$ as the time period from the beginning up to $t$. For example, $\mathcal{U}_{:t}$ denotes a set of items accumulated up to time period $t$ from the beginning (i.e., a set of entire users in the system).

**Dynamic recommendation.** We assume that an initial recommendation model has been pre-trained with $\mathcal{D}_0 = \{\mathcal{U}_0, \mathcal{I}_0, \mathcal{E}_0, \mathbf{Y}_0\}$ in an offline manner, and then the model is trained solely with the newly collected data $\mathcal{D}_t$ at the current time period $t$, $\forall t \in \{1, \ldots, T\}$. Once the model has been trained/fine-tuned on $\mathcal{D}_t$, a top-$K$ recommendation list $[i_1, \ldots, i_K]$ for each user $u$, ranked by the predicted scores $\widehat{\mathbf{Y}}_t[u, i], \forall i$, is generated.

**User-side fairness.** Given a binary sensitive attribute $a \in \{0, 1\}$ (e.g., gender), we focus on ensuring user-side group fairness, i.e., mitigate the recommendation performance disparity between the advantaged user group ($a = 0$) and the disadvantaged user group ($a = 1$) [17]. More specifically, the user-side performance disparity at any time period $t$ is defined as follows.

**DEFINITION 1** (User-side performance disparity [17]). *For a time period $t$ with ground-truth test interaction set $\mathcal{D}_t^{\text{test}}$ and for a recommendation metric* $\text{Perf}(\cdot)$ *(such as NDCG@$K$ or F1@$K$), the* user-side performance disparity *is defined by*

$$\text{PD}_t := \text{Perf}(\mathcal{D}_t^{\text{test}} \mid a = 0) - \text{Perf}(\mathcal{D}_t^{\text{test}} \mid a = 1). \quad (1)$$

**Problem definition.** We formally define the problem of dynamic user-side fairness in recommender systems as follows.

---

[1] Depending on the needs of the system or implementation, the time period could be either a specific time frame (e.g., daily, weekly, monthly) or until a specific number of interactions has been collected.

PROBLEM 1 (User-side fairness in dynamic recommender systems). **Input:** *(1) a pre-trained recommendation model with parameters* $\mathcal{W}_0$*; (2) a continually collected dataset* $\mathcal{D}_t = \{\mathcal{U}_t, \mathcal{I}_t, \mathcal{E}_t, Y_t\}$*,* $\forall t \in \{1, \ldots, T\}$*; (3) a binary sensitive attribute* $a \in \{0, 1\}$*; (4) a specific performance evaluation metric* Perf$(\cdot)$ *to calculate* $\mathrm{PD}_t$ *(see Eq. (1)).*

**Output:** *For any time period* $t$*, a fairness-regularized model with the parameters* $\mathcal{W}_t$ *that (1) optimizes the* $\mathrm{PD}_t$ *to be close to zero and (2) achieves high-quality recommendations.*

## 3 FADE: A FAIR DYNAMIC RECOMMENDER

In this section, we present FADE, a novel dynamic fair recommender system designed to effectively and efficiently reduce performance disparity over time. We begin with theoretical analyses on fine-tuning v.s. retraining in the context of dynamic fair recommendation in §3.1, demonstrating that the best practice is incremental fine-tuning with restart. Then in §3.2, we introduce our incremental fine-tuning strategy that balances both recommendation performance and user-side fairness. To address the non-differentiability challenge, we improve NeuralNDCG [25] and develop *Differentiable Hit* (DH), an efficient approximation scheme of the non-differentiable ranking metric, in §3.3. Building upon DH, we propose a differentiable and lightweight loss function for user-side fairness in §4.3. Our method is presented in Algorithm 1.

### 3.1 Fine-Tuning v.s. Retraining

Common practice for evolving data includes incremental *fine-tuning* and *retraining*. To obtain a deeper understanding of their behaviors in dynamic fair recommendation to guide the design of our method, we theoretically analyze their generalization error (recommendation & fairness) under distribution shift. Suppose that the model is currently trained with $\mathcal{D}_0 \cup \cdots \cup \mathcal{D}_{t_{te}-1}$ and is to be tested on $\mathcal{D}_{t_{te}}$. For each time period $t$, let $m_t := |\mathcal{E}_t|$ denote the size of dataset $\mathcal{D}_t$, let $\mathcal{L}^{\mathcal{D}_t}(\mathcal{W})$ denote the empirical loss (recommendation + fairness) over dataset $\mathcal{D}_t$, let $\mathcal{L}_t(\mathcal{W}) := \mathbb{E}_{\mathcal{D}_t}[\mathcal{L}^{\mathcal{D}_t}(\mathcal{W})]$ denote the true generalization loss, and let $\mathcal{L}_t^* := \inf_{\mathcal{W}} \mathcal{L}_t(\mathcal{W})$ denote the optimal loss value. To obtain concrete yet non-trivial theoretical results, we let $m_1 = \cdots = m_{t_{te}-1} \ll m_0$ and make mild and realistic assumptions for theoretical analysis (see §A.1).

Next, we introduce our theoretical measure of distribution shift. There are two sources of distribution shift over time: *covariate shift* and *concept drift*. In dynamic recommendation, covariate shift corresponds to shift of user/item attribute distributions (i.e., the distribution of $(\mathcal{U}_t, \mathcal{I}_t, \mathcal{E}_t)$), and concept drift corresponds to evolution of user preferences (i.e., the conditional distribution $Y_t | (\mathcal{U}_t, \mathcal{I}_t, \mathcal{E}_t)$).

Regarding covariate shift, a classic measure is the *discrepancy distance* [21] (a generalized $\mathcal{H}\Delta\mathcal{H}$ distance [2]):

$$d_{t,t_{te}}^{\mathcal{H}\Delta\mathcal{H}} := \sup_{\mathcal{W}, \mathcal{W}'} \left| |\mathcal{L}_t(\mathcal{W}) - \mathcal{L}_t(\mathcal{W}')| - |\mathcal{L}_{t_{te}}(\mathcal{W}) - \mathcal{L}_{t_{te}}(\mathcal{W}')| \right|. \quad (2)$$

The intuition is that if there is no covariate shift between $t$ and $t_{te}$, then for any two models $\mathcal{W}, \mathcal{W}'$, their difference of $\mathcal{L}$ should not differ between $t$ and $t_{te}$, leading to $d_{t,t_{te}}^{\mathcal{H}\Delta\mathcal{H}} = 0$. Regarding concept drift, we use a classic measure called *combined error* [2]:

$$d_{t,t_{te}}^{\mathrm{comb}} := \inf_{\mathcal{W}} \left( \mathcal{L}_t(\mathcal{W}) + \mathcal{L}_{t_{te}}(\mathcal{W}) \right) - \mathcal{L}_t^* - \mathcal{L}_{t_{te}}^*. \quad (3)$$

The intuition is that if there is no concept drift between $t$ and $t_{te}$, then $\mathcal{L}_t$ and $\mathcal{L}_{t_{te}}$ can achieve their minimum values with the same model $\mathcal{W}$, leading to $d_{t,t_{te}}^{\mathrm{comb}} = 0$. Together, we define a unified measure of distribution shift as follows by combining the measures of covariate shift and concept drift:

$$d_{t,t_{te}} := d_{t,t_{te}}^{\mathcal{H}\Delta\mathcal{H}} + d_{t,t_{te}}^{\mathrm{comb}}. \quad (4)$$

Building upon the measure of distribution shift, we theoretically analyze the overall behavior (recommendation performance & user-side fairness) of fine-tuning and retraining in the presence of distribution shift (Theorems 3.1 & 3.2).

THEOREM 3.1 (FINE-TUNING). *Let* $\mathcal{L}_{t_{te}}^{\mathrm{ft}}$ *denote the best possible loss of fine-tuning till* $\mathcal{D}_{t_{te}-1}$*. Suppose that the number of fine-tuning epochs at each time period* $t \geq 1$ *is decided according to the proximity assumption [27] with some* $0 < \gamma < 1$ *(see §A.1 for detail). Then with probability at least* $1 - \delta$*,*

$$\mathcal{L}_{t_{te}}^{\mathrm{ft}} \leq \mathcal{L}_{t_{te}}^* + \frac{(1-\gamma)\left(2 \sum_{t=0}^{t_{te}-1} \gamma^{t_{te}-t-1} d_{t,t_{te}} + 4\sqrt{\left(\frac{\gamma^{2t_{te}-2}}{m_0} + \frac{1-\gamma^{2t_{te}-2}}{(1-\gamma^2)m_1}\right)\log\frac{2}{\delta}}\right)}{1-\gamma^{t_{te}}}. \quad (5)$$

THEOREM 3.2 (RETRAINING). *Let* $\mathcal{L}_{t_{te}}^{\mathrm{rt}}$ *be the best possible loss of retraining on* $\mathcal{D}_0 \cup \cdots \cup \mathcal{D}_{t_{te}-1}$*. With probability at least* $1 - \delta$*,*

$$\mathcal{L}_{t_{te}}^{\mathrm{rt}} \leq \mathcal{L}_{t_{te}}^* + \frac{2m_0 d_{0T} + 2\sum_{t=1}^{t_{te}-1} m_1 d_{t,t_{te}}}{m_0 + (t_{te}-1)m_1} + 4\sqrt{\frac{1}{m_0 + (t_{te}-1)m_1}\log\frac{2}{\delta}}. \quad (6)$$

Proofs are in §A.2. Theorems 3.1 & 3.2 point out two sources of generalization error: (i) distribution shift in terms of $d_{t,t_{te}}$ and (ii) learning error due to the finite dataset size $m_t$. Regarding distribution shift, since larger time differences typically result in larger distribution shifts, we have $d_{0,t_{te}} > d_{1,t_{te}} > \cdots > d_{t_{te}-1,t_{te}}$ Fine-tuning can exponentially shrink (via the $\gamma^{t_{te}-t-1}$ factor) the influence of distribution shift while retraining suffers from greater influence of distribution shift. This is consistent with our intuition since retraining treats old and new data equally while fine-tuning pays more attention to newer data. This suggests that we should use fine-tuning to mitigate the impact of distribution shift. Meanwhile, when $t_{te}$ is large, fine-tuning's learning error $\frac{(1-\gamma)^2}{(1-\gamma^2)m_1}$ will be greater than retraining's $\frac{1}{m_0+(t_{te}-1)m_1}$ because $m_1 \ll m_0$. This suggests that the performance of dynamically fine-tuned model will eventually degrade after a number of periods, which is consistent with our empirical observation (refer to Fig. 10 in §B.3).

Therefore, to utilize the higher efficiency of fine-tuning without sacrificing performance, we propose to fine-tune the model for some periods $T$ until the performance starts to degrade. After that, we retrain the model from scratch and repeat the fine-tuning process again.

### 3.2 Incremental Fine-Tuning Strategy

Building upon our theoretical analysis on distribution shift and for the sake of time efficiency, FADE fine-tunes the model parameters incrementally over time only with the new data $\mathcal{D}_t$ collected at time period $t$. We optimize the following loss functions:

$$\mathcal{L}^{\mathcal{D}_t} := \mathcal{L}_{\mathrm{rec}}^{\mathcal{D}_t} + \lambda \mathcal{L}_{\mathrm{fair}}^{\mathcal{D}_t}, \quad (7)$$

where $\mathcal{L}_{\text{rec}}$ is for improving the recommendation performance, $\mathcal{L}_{\text{fair}}$ is for regularizing the performance disparity between the disadvantaged and advantaged groups, and $\lambda$ is the scaling parameter for controlling the trade-off between the recommendation performance and the fairness. In this paper, we use the classic Bayesian personalized ranking (BPR) loss [28] as $\mathcal{L}_{\text{rec}}$, i.e.,

$$\mathcal{L}_{\text{rec}}^{\mathcal{D}_t} := -\frac{1}{|\mathcal{E}_t|} \sum_{(u,i)\in\mathcal{E}_t} \frac{1}{|\mathcal{N}_{ui}|} \sum_{i'\in\mathcal{N}_{ui}} \log(\sigma(s_{ui} - s_{ui'})), \quad (8)$$

where $\sigma(\cdot)$ is the sigmoid function, and $\mathcal{N}_{ui}$ is a set of sampled negative items for $u$. Note that this loss can be replaced with any differentiable recommendation loss that can be optimized by gradient descent. We will define $\mathcal{L}_{\text{fair}}$ in §4.3.

By jointly optimizing $\mathcal{L}_{\text{rec}}$ and $\mathcal{L}_{\text{fair}}$ in an end-to-end fashion to fine-tune the model parameters for each time period, we can dynamically reduce the performance disparity, which may otherwise worsen as the optimization continues, while simultaneously accurately preserving the user preferences over time.

## 3.3 Differentiable Hit

Most evaluation metrics for top-$K$ recommendations, such as NDCG@$K$, are not differentiable due to their reliance on the ranking/sorting operation of items. As discussed in §1, this non-differentiability presents a challenge when optimizing fairness measures, specifically performance disparity, using gradient descent algorithms. To overcome this challenge, several soft ranking losses have been proposed to directly optimize relaxed ranking metrics [3, 25, 26]. NeuralNDCG [25] is a recent work on differentiable approximation of ranking metrics. However, due to the use of the Sinkhorn algorithm, NeuralNDCG is not only inefficient but also suffers from gradient vanishing issue. To address these limitations, we improve NeuralNDCG and propose *Differentiable Hit*, a function that is not only effective but also more lightweight than existing methods, making it well-suited for dynamic recommendation.

First, let us define a standard *Hit* function. Suppose a score vector $\mathbf{s}_u = [s_{u1}, s_{u2}, \ldots, s_{uN}]^\top$ for a user $u$ represents the *"unsorted"* list of recommendation scores (i.e., $s_{ui} = \hat{Y}_t[u,i]$) of $N$ *candidate items* in a set $C_u$ (with $|C_u| = N$), a vector $\mathbf{r}_u$ represents the *"sorted"* list of items ranked in the descending order by their scores in $\mathbf{s}_u$, and $\mathbf{r}_u[k]$ represents the $k$-th ranked item.

With the above definitions, we can define the Hit function, $\text{Hit}(C_u; k)$ for $k \in \{1, \ldots, K\}$, which indicates whether the $k$-th ranked item $\mathbf{r}_u[k]$ is $u$'s ground-truth item, as follows:

$$\text{Hit}(C_u; k) := \begin{cases} 1 & \text{if } Y_t[u, \mathbf{r}_u[k]] = 1, \\ 0 & \text{if } Y_t[u, \mathbf{r}_u[k]] = 0. \end{cases} \quad (9)$$

Here, the *sorting operation* used to produce the $\mathbf{r}_u$, which can also be represented as *a permutation matrix*, renders the $\text{Hit}_u(k)$ non-differentiable. However, we can overcome this limitation by using the continuous relaxation for permutation matrices to approximate the deterministic sorting operation with a differentiable continuous sorting [9]. First, for the deterministic sorting, the permutation matrix $\mathbf{P}_u \in \mathbb{R}^{N \times N}$ is given by [9]:

$$\mathbf{P}_u[k, j] := \begin{cases} 1 & \text{if } j = \arg\max[(N + 1 - 2k)\mathbf{s}_u - \mathbf{A}_u\mathbf{1}], \\ 0 & \text{otherwise,} \end{cases} \quad (10)$$

---

**Algorithm 1** Fine-tuning procedure at time period $t$

1: **Input:** Model parameters $\mathcal{W}_{t-1}$, scaling parameter $\lambda$, temperature parameter $\tau$, the number of negative items $n$ for $\mathcal{L}_{\text{rec}}$ and $\mu$ for $\mathcal{L}_{\text{fair}}$, sensitive attribute $a$, incoming dataset $\mathcal{D}_t = \{\mathcal{U}_t, \mathcal{I}_t, \mathcal{E}_t, Y_t\}$
2: **Output:** Updated model parameters $\mathcal{W}_t$
3: $\mathcal{W}_t \leftarrow \mathcal{W}_{t-1}$;
4: **for** epoch **do**
5:    **for** mini-batch $\mathcal{B}$ obtained from $\mathcal{E}_t$ **do**
6:       **for** user-item interaction $(u, i) \in \mathcal{B}$ **do**
7:          Sample $n$ negative items as $\mathcal{N}_{ui}$;
8:          Sample $\mu$ negative items as $\mathcal{N}'_{ui}$; $C_{ui} \leftarrow \{i\} \cup \mathcal{N}'_{ui}$;
9:       **end for**
10:      $\mathcal{L}_{\text{rec}} \leftarrow -\frac{1}{|\mathcal{B}|}\sum_{(u,i)\in\mathcal{B}}\frac{1}{|\mathcal{N}_{ui}|}\sum_{i'\in\mathcal{N}_{ui}}\log(\sigma(s_{ui}-s_{ui'}))$;
11:      $\text{DPD} \leftarrow \frac{\sum_{(u,i)\in\{\mathcal{B}|a=0\}}\text{DH}(C_{ui};1)}{|\{\mathcal{B}|a=0\}|} - \frac{\sum_{(u,i)\in\{\mathcal{B}|a=1\}}\text{DH}(C_{ui};1)}{|\{\mathcal{B}|a=1\}|}$;
12:      $\mathcal{L}_{\text{fair}} \leftarrow -\log(\sigma(-\text{DPD}))$;
13:      Update $\mathcal{W}_t$ based on $\mathcal{L}_{\text{rec}} + \lambda\mathcal{L}_{\text{fair}}$ via gradient descent;
14:    **end for**
15: **end for**
16: **return** $\mathcal{W}_t$;

---

where $\mathbf{1}$ is the column vector of all ones and $\mathbf{A}_u$ is the absolute distance matrix of $\mathbf{s}_u$ with $\mathbf{A}_u[k, j] = |s_{uk} - s_{uj}|$. For instance, if we set $k = \lfloor (N + 1)/2 \rfloor$, then the non-zero entry in the $k$-th row, $\mathbf{P}_u[k, :]$, corresponds to the element with the minimum sum of absolute distances to the other elements, and this corresponds to the median element, as desired.

Then, the argmax operator is replaced by Gumbel-softmax [12] to obtain a continuous relaxation of the permutation matrix; the $k$-th row of the permutation matrix is relaxed as follows [9]:

$$\widehat{\mathbf{P}}_u[k, :] := \text{softmax}\left[\left((N + 1 - 2k)\mathbf{s}_u - \mathbf{A}_u\mathbf{1}\right)/\tau\right], \quad (11)$$

where $\tau$ is the temperature parameter, and $\widehat{\mathbf{P}}_u$ approaches a permutation matrix (i.e., Eq. (10)) when $\tau \to 0^+$. Intuitively, each entry of $\widehat{\mathbf{P}}_u[k, :]$ indicates the probability that the corresponding item will be the $k$-th ranked item. Since $\widehat{\mathbf{P}}_u$ is continuous everywhere and differentiable almost everywhere w.r.t. the elements of $\mathbf{s}_u$, we can define a differentiable Hit, as we elaborate below.

Since the $k$-th row of the permutation matrix $\mathbf{P}_u[k, :]$ (i.e., Eq. (10)) is equal to the one-hot vector of the $k$-th ranked item, we can reformulate the Hit function (i.e., Eq. (9)) as follows:

$$\text{Hit}(C_u; k) = \mathbf{P}_u[k, :] \cdot Y_t[u, :]^\top, \quad (12)$$

where $Y_t[u, i] = 1$ if the item $i$ is a ground-truth item, and 0 otherwise. Finally, by replacing $\mathbf{P}_u[k, :]$ (Eq. (10)) with $\widehat{\mathbf{P}}_u[k, :]$ (Eq. (11)), we define a *Differentiable Hit* (DH) as follows:

$$\text{DH}(C_u; k) := \widehat{\mathbf{P}}_u[k, :] \cdot Y_t[u, :]^\top. \quad (13)$$

Using DH as a building block, we can differentiably approximate various top-$K$ recommendation metrics. For example,

$$\text{NDCG@}K \approx \frac{1}{|\mathcal{U}_t|} \sum_{u\in\mathcal{U}_t} \frac{1}{\text{maxDCG}(C_u)} \sum_{k=1}^{K} \frac{\text{DH}(C_u; k)}{\log_2(k+1)}, \quad (14)$$

where $\text{maxDCG}(C_u)$ is the maximum possible value of $\sum_{k=1}^{K}\frac{\text{DH}(C_u;k)}{\log_2(k+1)}$, computed by decreasingly ordering $i \in C_u$ by $Y_t[u, i]$.

## 3.4 Fairness Loss

We design our fairness loss for reducing performance disparity between the advantaged ($a = 0$) and disadvantaged ($a = 1$) user groups. For the sake of training efficiency, we compose each candidate set with only 1 positive item and several negative items and use differentiable Hit@1 in our fairness loss. Formally, for each $(u, i) \in \mathcal{E}_t$, we sample $\mu$ negative items $\mathcal{N}'_{ui}$, compose a candidate set $C_{ui} := \{i\} \cup \mathcal{N}'_{ui}$ and use DH($C_{ui}$; 1) as a surrogate of the measure of recommendation quality for a user. While this differentiable Hit@1 used for training encourages the top-1 recommendation, it could also potentially benefit Hit@$K$-based metrics. We will empirically demonstrate that these settings consistently yield effective results across various recommendation metrics that rely on the Hit function. Based on DH, we define the differentiable performance disparity (DPD) as follows:

$$\text{DPD}^{\mathcal{D}_t} := \frac{\sum_{(u,i) \in \{\mathcal{E}_t | a=0\}} \text{DH}(C_{ui}; 1)}{|\{\mathcal{E}_t | a = 0\}|} - \frac{\sum_{(u,i) \in \{\mathcal{E}_t | a=1\}} \text{DH}(C_{ui}; 1)}{|\{\mathcal{E}_t | a = 1\}|}, \quad (15)$$

which is an approximation of $\text{PD}_t$ in Eq. (1) on the sampled item set. Then, a naïve fairness loss function is to minimize |DPD|:

$$\mathcal{L}^{\mathcal{D}_t}_{\text{fair-abs}} := -\log(\sigma(-|\text{DPD}^{\mathcal{D}_t}|)), \quad (16)$$

where $\sigma(\cdot)$ is the sigmoid function. However, the non-smoothness of $\mathcal{L}_{\text{fair-abs}}$ will cause instability in training, as shown in our experiment (Fig. 2). To address this limitation, we leverage the property of the sigmoid function and surprisingly prove that removing the absolute value operation $|\cdot|$ can still ensure fairness adaptively. Formally, we propose the following fairness loss:

$$\mathcal{L}^{\mathcal{D}_t}_{\text{fair}} := -\log(\sigma(-\text{DPD}^{\mathcal{D}_t})). \quad (17)$$

Then we have Proposition 3.3.

PROPOSITION 3.3. *Let* $\widetilde{\mathcal{W}_t} := \mathcal{W}_t - \eta \nabla_{\mathcal{W}_t}(\mathcal{L}^{\mathcal{D}_t}_{\text{rec}} + \lambda \mathcal{L}^{\mathcal{D}_t}_{\text{fair}})$ *denote a gradient descent step with learning rate* $\eta > 0$. *Suppose that* $\mathcal{L}_{\text{rec}}$ *causes unfairness (i.e.,* $\langle \nabla_{\mathcal{W}_t} \mathcal{L}^{\mathcal{D}_t}_{\text{rec}}, \nabla_{\mathcal{W}_t} \text{DPD}^{\mathcal{D}_t} \rangle \leq 0$*), and that the fairness loss has influence (i.e.,* $\nabla_{\mathcal{W}_t} \mathcal{L}^{\mathcal{D}_t}_{\text{fair}} \neq \mathbf{0}$*). Then, there exists* $\lambda \geq 0$ *such that*

$$\text{sgn}(\text{DPD}^{\mathcal{D}_t}(\mathcal{W}_t)) \cdot \lim_{\eta \to +0} \frac{\text{DPD}^{\mathcal{D}_t}(\widetilde{\mathcal{W}_t}) - \text{DPD}^{\mathcal{D}_t}(\mathcal{W}_t)}{\eta} \leq 0. \quad (18)$$

*In particular, if* $\text{DPD}^{\mathcal{D}_t}(\mathcal{W}_t) \leq 0$*, then* $\text{DPD}^{\mathcal{D}_t}(\widetilde{\mathcal{W}_t}) \geq \text{DPD}^{\mathcal{D}_t}(\mathcal{W}_t)$ *as* $\eta \to +0$.

Proof is in §A.3. Intuitively, our $\mathcal{L}_{\text{fair}}$ aims to benefit the disadvantaged user group ($a = 1$) over the advantaged group ($a = 0$). Meanwhile, whenever DPD < 0, the influence of $\mathcal{L}_{\text{fair}}$ will be reduced adaptively, so the unfair $\mathcal{L}_{\text{rec}}$ will push DPD back to zero.

## 3.5 Complexity Analysis

Our fairness loss only adds a constant amount of complexity to most existing recommendation models. Assuming we employ MF-BPR [28] as the base recommendation model with user/item embeddings of dimensionality $d$, the time complexity of minimizing $\mathcal{L}^{\mathcal{D}_t}_{\text{rec}}$ is $O(|\mathcal{E}_t|nd)$, where $n$ represents the number of negative items.

Regarding our fairness loss, *for each user interaction*, computing the score vector $\mathbf{s}_u$ has a time complexity of $O(\mu d)$, and computing DH incurs $O(\mu^2)$ time complexity due to computing $\widehat{\mathbf{P}}_u[k, :]$ (i.e.,

Eq. (11)), which involves computing $\mathbf{A}_u \in \mathbb{R}^{(\mu+1) \times (\mu+1)}$. As a result, the time complexity of minimizing $\mathcal{L}^{\mathcal{D}_t}_{\text{fair}}$ becomes $O(|\mathcal{E}_t|(\mu^2 + \mu d))$, which can be approximated as $O(|\mathcal{E}_t|\mu d)$ since $\mu \ll d$. Therefore, the time complexity of minimizing the recommendation loss, $\mathcal{L}^{\mathcal{D}_t}_{\text{rec}}$, and the fairness loss, $\mathcal{L}^{\mathcal{D}_t}_{\text{fair}}$, are comparable.

## 4 EXPERIMENTS

We design experiments to answer the following key research questions (RQs)[2]:

**RQ1.** How does learning new data affect model overall behavior?
**RQ2.** How effective is the fairness loss and fine-tuning in FADE?
**RQ3.** Does FADE outperform its fairness-aware competitors?
**RQ4.** How time-efficient is FADE?
**RQ5.** How effective/efficient is the Differentiable Hit in FADE?
**RQ6.** How sensitive is FADE to its hyperparameters?

## 4.1 Experimental Settings

*4.1.1 Dataset.* For experiments, we use two real-world recommendation datasets from different domains.

- Movielens[3]: This dataset contains $836, 478$ ratings on $3, 628$ movies by $6, 039$ users at different timestamps. The sensitive attribute $a$ is determined by the gender of each user, with male users as $a = 0$ (advantaged) and female users as $a = 1$ (disadvantaged). This classification is based on the observation that the dataset is male-dominated, consisting of $4, 330$ male users with $627, 933$ training instances and $1, 709$ female users with $208, 545$ training instances [19].
- ModCloth[4] [32]: This e-commerce dataset contains $83, 147$ ratings on $1, 014$ items (i.e., women's clothing) by $37, 142$ users at different timestamps. The sensitive attribute $a$ is determined by the body shape of each user, with "Small" users as $a = 0$ (advantaged) and "Large" users as $a = 1$ (disadvantaged). The dataset is dominated by "Small" users, comprising $28, 374$ "Small" users with $66, 663$ training instances and $8, 768$ "Large" users with $16, 484$ training instances.

Following previous works in recommender systems [13, 39], we binarize the 5-star ratings for both datasets. We set $\mathbf{Y}[u, i] = 1$ if user $u$ gives item $i$ a rating greater than 2, and $\mathbf{Y}[u, i] = 0$ otherwise. Note that the dataset descriptions provided earlier are based on these pre-processed datasets.

To simulate dynamic settings defined in §2, we first sort the interactions in the dataset in chronological order and use 60%/70% of them as pre-training data, and 28%/21% as dynamically observed data for Movielenz/ModCloth. We then split the dynamically observed data into 7 periods, each containing an equal number of interactions. This process yields $\{\mathcal{D}_0, \mathcal{D}_1, \ldots, \mathcal{D}_T\}$, where $T = 7$.

*4.1.2 Compared methods.* To ensure that the effectiveness of FADE is independent of the base recommender system used, we use two base system including Matrix Factorization (MF) and Neural Collaborative Filtering (NCF), both with the Bayesian Personalized Ranking (BPR) loss [28]. In this setup, we aim to validate the effectiveness of our *fine-tuning strategy* and the *fairness loss* used in

---

[2]Note that throughout the subsections for all RQs, we use PD to refer to absolute performance disparity |PD|.
[3]https://grouplens.org/datasets/movielens/1m/
[4]https://github.com/MengtingWan/marketBias

FADE in ensuring high recommendation performance and user-side fairness over time. To establish a benchmark, we compare FADE with the following six combinations:

- PRETRAIN/PRETRAIN-FAIR: The static model pre-trained on $\mathcal{D}_0$ w/o and w/ the fairness loss, respectively.
- RETRAIN/RETRAIN-FAIR: Fully retraining the model using the accumulated historical data $\mathcal{D}_{:t}$ at each time period $t$, w/o and w/ the fairness loss, respectively.
- FINETUNE/FADE-ABS: Fine-tuning the model based on the current $\mathcal{D}_t$ at each $t$, w/o the fairness loss and w/ the (naïve) fairness loss $\mathcal{L}_{\text{fair-abs}}$ that uses |DPD| (Eq. 16), respectively.

In addition, we also compare FADE with the other fairness-aware competitors. To ensure a fair comparison, we implemented these methods with a fine-tuning strategy, even though they were originally not based on fine-tuning. The competitors we consider are:

- ADVER [18]: This method is based on adversarial learning technique. It is originally designed to filter out sensitive attributes from user embeddings, but its primary focus is not on reducing the performance disparity among different user groups.
- Re-rank [17]: This method is a fairness-constrained re-ranking approach. At each time period, a fine-tuned base model generates recommendation lists, which are used as the basis for generating new fair recommendation lists using this method.

*4.1.3 Evaluation tasks.* To evaluate the recommendation recommendation performance and the disparity, we design two types of recommendation tasks:

- Task-Remain (Task-R): Given the model trained up until time period $t$, the model is tested by recommending items for the remaining time periods with the test set $\mathcal{D}_t^{\text{test}} = \mathcal{D}_{t+1} \cup \cdots \cup \mathcal{D}_T$.
- Task-Next (Task-N): Given the model trained up until time period $t$, the model is tested by recommending items for the right-next time period with the test set $\mathcal{D}_t^{\text{test}} = \mathcal{D}_{t+1}$.

Note that for both tasks, the data at the last time period, $\mathcal{D}_T$, is only used for testing and not for training purposes. Due to space issue, we put the full results for Task-N in §B.3.

We use widely-used metrics normalized discounted accumulated gain 20 (NDCG@20) and F1@20 to evaluate the top-20 recommendation quality. We adopt a similar approach as previous studies [14, 17], where we randomly sample 100 items that the user has not interacted with as negative samples. These negative samples, along with the ground-truth items, are used for evaluation.

*4.1.4 Implementation details.* For all compared methods, we set $n$ (the number of negative samples for BPR loss) to 4, the learning rate to 0.001, and L2 regularization to 0.0001. We use the Adam optimization algorithm [15] to update model parameters.

For FADE and RETRAIN-FAIR based on both MF and NCF, we set $\tau = 3$, $\mu = 4$, and the number of dynamic update epochs to 10, which consistently show excellent trade-off between performance and disparity across all metrics, tasks, and datasets. The $\lambda$ is selected within range [0, 4] for PRETRAIN-FAIR, RETRAIN-FAIR, FADE-ABS, and FADE in all cases. We use a random seed for better reproducibility. For the implementation details of RERANK [17] and ADVER [18], please refer to §B.1

## 4.2 The Effect of Learning from New Data

For RQ1 and RQ2, we compare the recommendation performance and performance disparity, both averaged across each dynamic update data, of the five methods (PRETRAIN, RETRAIN, FINETUNE, PRETRAIN-FAIR, RETRAIN-FAIR) with FADE. Fig. 2 shows the results w.r.t. different metrics, base recommender, and datasets.

First, compared to PRETRAIN, RETRAIN and FINETUNE yield an average increase of 9.01% and 4.61%, respectively, in recommendation performance in *all* cases, indicating that the new data is indeed useful for improving recommendation performance of the models over time. For PRETRAIN-FAIR, RETRAIN-FAIR, and FADE, the similar trend is observed: an average increase of 4.66% and 4.09%, respectively. However, in some cases on ModCloth, FADE performs worse than PRETRAIN-FAIR due to the initial high disparity of PRETRAIN-FAIR.

Regarding performance disparity, the PDs of RETRAIN tend to exceed those of PRETRAIN, and those of FINETUNE tend to fall below but still remain significant. This highlights the need to incorporate fairness considerations when integrating new data.

## 4.3 Ablation Study of FADE

*4.3.1 With and without fairness loss.* To answer RQ2, we continue comparing FADE with aforementioned five methods. First, regarding disparity, Fig. 2 shows that RETRAIN-FAIR and FADE yield significantly lower PDs compared to RETRAIN and FINETUNE, in *all* cases, with an average reduction of 47.60% and 48.91%, respectively. The results indicate that our fairness loss indeed helps reduce the performance disparity at each time period.

Furthermore, we examine how disparities change over time with FADE and the three methods, RETRAIN, RETRAIN-FAIR, FINETUNE, as shown in Fig. 3. We can see that without the fairness loss (RETRAIN/FINETUNE), the PDs tend to persist relatively high over time in all cases. However, when augmented with the fairness loss (RETRAIN-FAIR/FADE), the PDs tend to remain stably low.

Besides significant reduction of PDs, FADEhas merely marginal sacrifice (2.44% on average) in recommendation performance compared to FINETUNE, and similar results are observed for RETRAIN and RETRAIN-FAIR, with an average decrease of 0.495%. This relatively slight decrease in recommendation performance is because FADE improves the performance of the disadvantaged group while reducing the performance of the advantaged group, in *all* cases, with an average increase of 2.06% and decrease of 3.37%, respectively.

*4.3.2 Fine-tuning v.s. Retraining.* Fig. 2 shows that FINETUNE consistently outperform RETRAIN w.r.t. both PD (an average decrease of 14.79%) and recommendation performance (an average increase of 1.38%) in all cases. FADE outperform RETRAIN-FAIR w.r.t. PD (an average decrease of 16.47%) while only slightly compromising recommendation performance (an average decrease of 0.61%). These results are consistent with our theoretical findings in §3.1, indicating that retraining is more affected by distribution shifts, while fine-tuning can exponentially shrink this impact. The lack of a clear advantage for fine-tuned models in recommendation performance is due to their eventual degradation after multiple periods, which is shown, for example, in the results for Movielenz in Fig. 8 in §B.3.

## 4.4 Comparison with Fairness Competitors

To answer RQ3, we further compare FADE with the two fairness-aware competitors, ADVER and RERANK, in Fig. 2. Note that all of

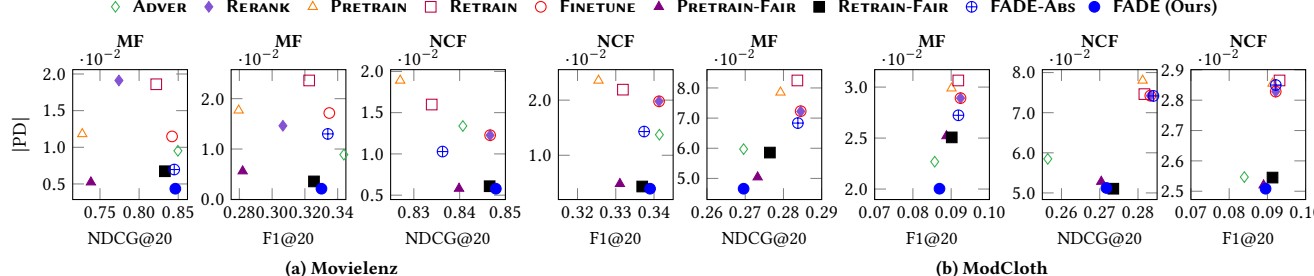

**(a) Movielenz** **(b) ModCloth**

**Figure 2: The trade-off between recommendation performance (NDCG@20 & F1@20) and absolute performance disparity |PD| of eight compared methods and FADE in Task-R. Employing our fairness loss leads to a substantial reduction in |PD| across all cases, with a modest impact on overall performance. Note that the optimal point should be situated in the bottom-right corner.**

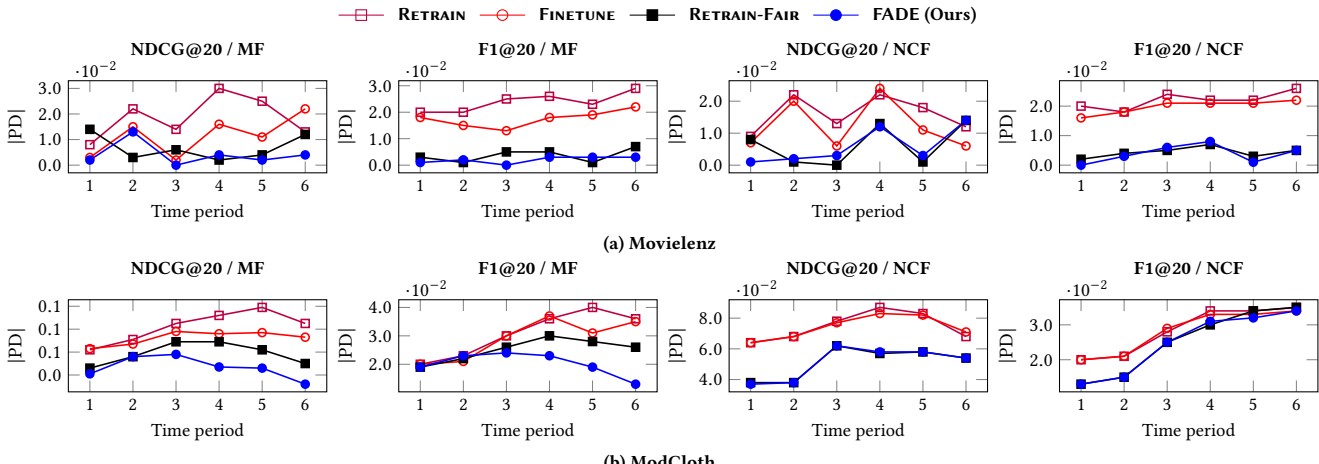

**(a) Movielenz**

**(b) ModCloth**

**Figure 3: The trend of the absolute performance disparity (|PD|) in Task-R. Without the fairness loss, the |PD| is relatively high and often increase, while with the fairness loss, particularly in FADE, the |PD| tends to remain relatively low.**

those methods are implemented based on fine-tuning strategy for fair comparison. First, FADE consistently achieves smaller PDs, averaging 36.53%, and it offers comparable recommendation performance on average 1.49% better than Adver. This is because Adver is not designed to reduce the performance gap between user groups; instead, its focus is on removing information related to sensitive attributes from user representations.

Rerank and Finetune yield similar results in many cases, meaning that its re-ranking algorithm struggle to effectively re-rank the given recommendation lists. This is because the given base recommendation lists are already too unfair. For example, for disadvantaged users, the predicted scores may not accurately reflect the user's true interests, resulting in very low predicted scores for the ground-truth items in the list. This issue is exacerbated when the given recommendation lists are short, which is a common in practice. This observation agrees with our intuition that dynamic adaptation is necessary rather than using post-processing.

## 4.5 Time-efficiency Comparison

To answer RQ4, we compare running time of FADE with the full-retraining based methods and the other fairness-aware techniques. The results are in Table 2 and each entry is the average running time of a model across the dynamic update data at each time period.

We have several observations based on the running time, averaged over base models and datasets. Firstly, Finetune/FADE achieve

approximately 323/270 times faster running time compared to Retrain/Retrain-Fair, indicating that the fine-tuning strategy employed in FADE enables the models to achieve high time efficiency, making them ideal for dynamic settings. Secondly, Retrain-Fair/FADE exhibit approximately 1.06/1.27 times slower running time in comparison to Retrain/Finetune. This suggests that the additional computational cost introduced by our fairness loss is not significant. Lastly, FADE demonstrates a time efficiency around 10.23 times and 94.11 times faster than Adver and Rerank, respectively, highlighting the lightweight design of our fairness loss compared to the existing fairness-aware losses.

## 4.6 Comparison with Soft Ranking Metrics

Due to the space limit, the results for RQ5 are deferred to §B.4. In essence, they show that FADE outperforms or matches the variant of FADE adapting NeuralNDCG in both recommendation performance and disparity, while being approximately four times faster. This is because our differentiable Hit addresses NeuralNDCG's gradient vanishing issue by eliminating Sinkhorn's algorithm.

## 4.7 Hyperparameter Analysis

For RQ6, we investigate the sensitivity of FADE to four hyperparameters: (1) the scaling parameter $\lambda$, (2) the number of epochs of dynamic updates, (3) the temperature parameter $\tau$, and (4) the number of negative items $\mu$. Due to the space limit, we only show

**Table 2: Efficiency comparison on the running time (seconds).**

| Data | Models | Full-retrain-based | | Fine-tune-based | | | |
|------|--------|---------|-------------|-------|--------|----------|------|
| | | RETRAIN | RETRAIN-FAIR | ADVER | RERANK | FINETUNE | FADE |
| Movie. | MF | 1373.17 | 1401.18 | 55.16 | 132.46 | 2.57 | 4.08 |
| | NCF | 1381.59 | 1488.5 | 61.66 | 420.54 | 5.07 | 5.93 |
| Mod. | MF | 154.22 | 163.12 | 4.01 | 250.75 | 0.79 | 0.93 |
| | NCF | 188.58 | 242.29 | 4.01 | 344.51 | 1.15 | 1.26 |
| Average | | 774.39 | 823.77 | 31.21 | 287.06 | 2.40 | 3.05 |

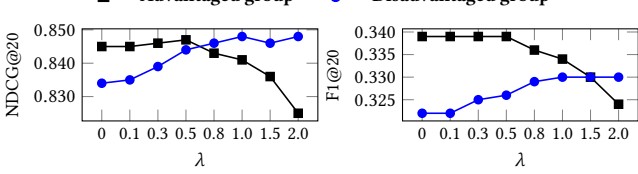

**Figure 4: The effect of the scaling parameter $\lambda$ on the performance of the advantaged and disadvantaged groups.**

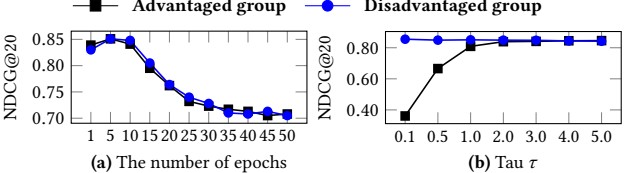

(a) The number of epochs          (b) Tau $\tau$

**Figure 5: Effect of hyperparamters.**

the results of FADE based on MF on Movielenz for $\lambda$, the number of epochs, and $\tau$ in Figs. 4 & 5. Please refer to §B.5 for the full results. They illustrate the performance of the advantaged and disadvantaged user groups for different values of these hyperparameters.

*4.7.1 Effect of scaling parameter $\lambda$ for the fairness Loss.* Fig. 4 shows that the performance of the advantaged group tend to decrease while that of the disadvantaged group tend to increase as $\lambda$ increases. In other words, the performance disparity between the two user groups steadily reduces until $\lambda$ reaches an optimal value, which varies depending on the specific metric used. The results indicate that $\lambda$ effectively controls the trade-off between recommendation performance and performance disparity.

*4.7.2 Effect of the number of epochs of dynamic updates.* Fig. 5-(a) shows that the performance of both user groups increases as the number of epochs of dynamic fine-tuning increases until reaching a peak around epoch 5 or 10. Subsequently, the performance gradually declines with further increases in the number of epochs. We suspect that setting the number of epochs too low may result in the model not learning enough from the current data. Conversely, when the number of epochs is set too high, the model potentially loses the knowledge acquired from historical data. We argue that this phenomenon is well-suited for the dynamic environment, as setting a low value for the number of epochs results in high efficiency.

*4.7.3 Effect of temperature parameter $\tau$ in the relaxed permutation matrix.* Higher values of $\tau$ result in smoother rows in the relaxed permutation matrix, $\widehat{\mathbf{P}}_u[i,:]$. Fig 5-(b) shows that the performance of both user groups increases until $\tau = 2$, and then stabilizes. These findings indicate that FADE is not highly sensitive to $\tau$, consistently delivering excellent performance for both user groups as long as $\tau$ is not too small. When $\tau$ is set too low, the Gumbel-softmax distribution becomes sharp, resulting in a nearly deterministic decision-making process for the model, i.e., $\widehat{\mathbf{P}}_u[i,:]$ will be

close to the one-hot vector of the $i$-th ranked item. As a result, the entry corresponding to the positive item in that vector is likely to have an extremely small value, from the initial phase of training, potentially hindering the the fairness regularization.

## 5 RELATED WORK

**Dynamic recommender systems.** Instead of fully retraining with the entire dataset when new data is collected, which can be time-inefficient, we can fine-tune the model parameters using only the new data, which is referred to as dynamic/online recommender systems in the literature. To effectively learn from relatively sparse new data, several methods have been proposed based on reweighting either (1) the impact of each user-item interaction [10, 29] or (2) that of each model parameter [6, 20, 38]; [14] utilizes both approaches. One unique advantage of the fairness loss in FADE is that it can be easily applied to any existing dynamic recommender systems optimized using gradient-based algorithms.

**Fair recommender systems in dynamic scenarios.** Various fairness demands exist in recommender systems, including user-side, item-side, and multi-side fairness. User-side fairness ensures fair recommendation quality for different users, while item-side fairness concentrates on equal exposure opportunities for items in recommendations. Two-sided fairness seeks to balance these two aspects. While the literature [8, 23, 40] has addressed item-side fairness in dynamic recommendations, such as the work by [40] that scales predicted ratings by item popularity with higher strength over time, user-side fairness in dynamic settings remains unexplored, to the best of our knowledge.

As described in Section 1, existing user-side fairness-aware re-ranking methods [7, 17] face the difficulties in dynamic settings. These methods tend to be time-inefficient, involving optimization problem akin to 0-1 integer programming problem. Furthermore, their non-differentiable fairness constraint, separating fairness optimization from that of recommendation quality, precludes model parameters from being regularized by fairness constraints. This hinders adaptation to distribution shifts in dynamic settings.

Another line of research into user-side fairness [1, 34, 35] employs adversarial functions to generate fair user representations independent of sensitive user attributes. However, these formulations do not explicitly address the reduction of performance disparity.

## 6 CONCLUSION

In this paper, we study the problem of user-side fairness in the dynamic recommendation scenario. We point out three key challenges in this problem: (1) distribution shifts, (2) frequent model updates, and (3) non-differentiability of ranking metrics. To address these challenges, we begin with theoretical analyses on fine-tuning v.s. retraining, showing that the best practice is incremental fine-tuning with restart. Guided by these insights, we propose FAir Dynamic rEcommender (FADE), an end-to-end fine-tuning framework that dynamically ensures user-side fairness over time. It incorporates our fairness loss equipped with our lightweight Differentiable Hit, which alleviating the gradient vanishing issue in the recent Neural-NDCG method and enhances efficiency. Through extensive experiments, we verify that FADE effectively and efficiently alleviates the performance disparity without significantly sacrificing recommendation performance.

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

# A  THEORETICAL ANALYSES

## A.1  Assumptions

In this subsection, we introduce our theoretical assumptions, which are quite mild and realistic.

To ensure that the dataset has a good coverage of the underlying distribution, a common assumption in literature is independence:

**ASSUMPTION 1** (DATA INDEPENDENCE). *For every $t$, the data tuples in $\mathcal{D}_t$ are mutually independent.*

Regarding the loss function, a well-behaved loss function should be able to be minimized. Common loss functions satisfy this property. This leads us to the following Assumption 2:

**ASSUMPTION 2** (EXISTENCE OF INFIMA). *For every $t$, the infimum $\mathcal{L}_t^* := \inf_{\mathcal{W}} \mathcal{L}_t(\mathcal{W})$ exists.*

Note that we do not assume *realizability*, i.e., we do not assume that there exists $\mathcal{W}$ that can achieve this infimum. Our Assumption 2 is realistic in machine learning. For example, neural networks can arbitrarily approximate any continuous function over any compact domain [24], but they may not be exactly equal that function.

Besides that, since data tuples are mutually independent, each data tuple in the dataset should not have dominant influence on the overall loss function, which means that the loss function should use the whole dataset. This leads us to the following Assumption 3:

**ASSUMPTION 3** (NO DOMINANT INFLUENCE). *For every $t$, for each data tuple $z \in \mathcal{D}_t$, the loss $\mathcal{L}^{\mathcal{D}_t}(\mathcal{W})$ conditioned on $\mathcal{D}_t \setminus \{z\}$ is $\frac{\varsigma^2}{m_t}$-subgaussian. Without loss of generality, we can assume $\varsigma = 1$ by rescaling $\mathcal{L}$.*

The subgaussian property is a common assumption in machine learning [22], and common loss functions satisfy our Assumption 3. Since there exist various definitions of the subgaussian property (yet equivalent up to constant factors), we clarify our definition as follows:

**DEFINITION 1** (SUBGAUSSIAN PROPERTY). *For $\varsigma > 0$, a real-valued random variable $X$ is said to be $\varsigma^2$-subgaussian if*

$$\mathbb{E}[e^{v(X-\mathbb{E}[X])}] \leq e^{\varsigma^2 v^2/2}, \quad \forall v \in \mathbb{R}. \tag{19}$$

*The equality holds for univariate Gaussians with variance $\varsigma^2$.*

Finally, we state our assumption on fine-tuning and retraining. For each $t \geq 1$, let $\mathcal{W}_t^{\text{ft}}$ denote the model parameters fine-tuned till $\mathcal{D}_t$. To characterize the fact that the fine-tuned $\mathcal{W}_t^{\text{ft}}$ does not completely forget the previously learned knowledge in $\mathcal{W}_{t-1}^{\text{ft}}$, we assume that all time periods share the same parameter space and use the following classic Assumption 4 (adapted from [27]):

**ASSUMPTION 4** (PROXIMAL FINE-TUNING). *There is $0 < \gamma < 1$ such that for each $t \geq 1$, the number of fine-tuning epochs is decided such that the fine-tuned $\mathcal{W}_t^{\text{ft}}$ is minimizing*

$$\ell_t(\mathcal{W}) := \mathcal{L}^{\mathcal{D}_t}(\mathcal{W}) + \gamma \ell_{t-1}(\mathcal{W}), \tag{20}$$

*where $\ell_0(\mathcal{W}) := \mathcal{L}^{\mathcal{D}_0}(\mathcal{W})$ denotes the pretraining loss function.*

For retraining, we assume that the influence of each time period $t$ to the retraining loss is a proportional to the size $m_t$ of $\mathcal{D}_t$:

**ASSUMPTION 5** (RETRAINING LOSS).

$$\mathcal{L}_{t_{\text{te}}-1}^{\text{rt}}(\mathcal{W}) := \frac{\sum_{t=0}^{t_{\text{te}}-1} m_t \mathcal{L}^{\mathcal{D}_t}(\mathcal{W})}{\sum_{t=0}^{t_{\text{te}}-1} m_t}. \tag{21}$$

Although this is a simplification of the retraining loss in practice, it still captures the essential properties of retraining.

## A.2  Proofs of Theorems 3.1 & 3.2

Our proofs of Theorems 3.1 & 3.2 rely the following Lemma A.1.

**LEMMA A.1.** *For $\boldsymbol{\alpha} \in \mathbb{R}_{\geq 0}^{t_{\text{te}}}$ with $\sum_{t=0}^{t_{\text{te}}-1} \alpha_t = 1$ and for $\epsilon > 0$, let $\mathcal{W}_{t_{\text{te}}-1}^{\boldsymbol{\alpha},\epsilon}$ denote some model parameters such that*

$$\sum_{t=0}^{t_{\text{te}}-1} \alpha_t \mathcal{L}_t(\mathcal{W}_{t_{\text{te}}-1}^{\boldsymbol{\alpha},\epsilon}) \leq \epsilon + \inf_{\mathcal{W}} \sum_{t=0}^{t_{\text{te}}-1} \alpha_t \mathcal{L}^{\mathcal{D}_t}(\mathcal{W}). \tag{22}$$

*Then with probability at least $1 - \delta$,*

$$\mathcal{L}_{t_{\text{te}}}(\mathcal{W}_{t_{\text{te}}-1}^{\boldsymbol{\alpha},\epsilon}) \leq \mathcal{L}_{t_{\text{te}}}^* + \epsilon + 2 \sum_{t=0}^{t_{\text{te}}-1} \alpha_t d_{t,t_{\text{te}}} + 4 \sqrt{\sum_{t=0}^{t_{\text{te}}-1} \frac{\alpha_t^2}{m_t} \log \frac{2}{\delta}}.$$

**PROOF OF LEMMA A.1.** Generalized from [2]. For $k \geq 1$, let

$$\mathcal{W}_t^k \in \mathcal{L}_t^{-1}\left((-\infty, \mathcal{L}_t^* + \tfrac{1}{k}]\right), \tag{23}$$

$$\mathcal{W}_{t,t_{\text{te}}}^k \in (\mathcal{L}_t + \mathcal{L}_{t_{\text{te}}})^{-1}\left((-\infty, \mathcal{L}_t^* + \mathcal{L}_{t_{\text{te}}}^* + d_{t,t_{\text{te}}}^{\text{comb}} + \tfrac{1}{k}]\right). \tag{24}$$

Then for any $\mathcal{W}$, by the triangle inequality,

$$\left| (\mathcal{L}_t(\mathcal{W}) - \mathcal{L}_t^*) - (\mathcal{L}_{t_{\text{te}}}(\mathcal{W}) - \mathcal{L}_{t_{\text{te}}}^*) \right| \tag{25}$$

$$= \left| |\mathcal{L}_t(\mathcal{W}) - \mathcal{L}_t^*| - |\mathcal{L}_{t_{\text{te}}}(\mathcal{W}) - \mathcal{L}_{t_{\text{te}}}^*| \right| \tag{26}$$

$$= \left| (|\mathcal{L}_t(\mathcal{W}) - \mathcal{L}_t(\mathcal{W}_{t,t_{\text{te}}}^k)| - |\mathcal{L}_T(\mathcal{W}) - \mathcal{L}_{t_{\text{te}}}(\mathcal{W}_{t,t_{\text{te}}}^k)|) \right.$$
$$+ (|\mathcal{L}_t(\mathcal{W}) - \mathcal{L}_t^*| - |\mathcal{L}_t(\mathcal{W}) - \mathcal{L}_t(\mathcal{W}_{t,t_{\text{te}}}^k)|)$$
$$\left. - (|\mathcal{L}_{t_{\text{te}}}(\mathcal{W}) - \mathcal{L}_{t_{\text{te}}}^*| - |\mathcal{L}_{t_{\text{te}}}(\mathcal{W}) - \mathcal{L}_{t_{\text{te}}}(\mathcal{W}_{t,t_{\text{te}}}^k)|) \right| \tag{27}$$

$$\leq \left| |\mathcal{L}_t(\mathcal{W}) - \mathcal{L}_t(\mathcal{W}_{t,t_{\text{te}}}^k)| - |\mathcal{L}_{t_{\text{te}}}(\mathcal{W}) - \mathcal{L}_{t_{\text{te}}}(\mathcal{W}_{t,t_{\text{te}}}^k)| \right|$$
$$+ \left| |\mathcal{L}_t(\mathcal{W}) - \mathcal{L}_t^*| - |\mathcal{L}_t(\mathcal{W}) - \mathcal{L}_t(\mathcal{W}_{t,t_{\text{te}}}^k)| \right|$$
$$+ \left| |\mathcal{L}_{t_{\text{te}}}(\mathcal{W}) - \mathcal{L}_{t_{\text{te}}}^*| - |\mathcal{L}_{t_{\text{te}}}(\mathcal{W}) - \mathcal{L}_{t_{\text{te}}}(\mathcal{W}_{t,t_{\text{te}}}^k)| \right| \tag{28}$$

$$\leq d_{t,t_{\text{te}}}^{\mathcal{H}\Delta\mathcal{H}} + \left| (\mathcal{L}_t(\mathcal{W}) - \mathcal{L}_t^*) - (\mathcal{L}_t(\mathcal{W}) - \mathcal{L}_t(\mathcal{W}_{t,t_{\text{te}}}^k)) \right|$$
$$+ \left| (\mathcal{L}_{t_{\text{te}}}(\mathcal{W}) - \mathcal{L}_{t_{\text{te}}}^*) - (\mathcal{L}_{t_{\text{te}}}(\mathcal{W}) - \mathcal{L}_{t_{\text{te}}}(\mathcal{W}_{tT}^k)) \right| \tag{29}$$

$$= d_{t,t_{\text{te}}}^{\mathcal{H}\Delta\mathcal{H}} + |\mathcal{L}_t(\mathcal{W}_{t,t_{\text{te}}}^k) - \mathcal{L}_t^*| + |\mathcal{L}_{t_{\text{te}}}(\mathcal{W}_{t,t_{\text{te}}}^k) - \mathcal{L}_{t_{\text{te}}}^*| \tag{30}$$

$$= d_{t,t_{\text{te}}}^{\mathcal{H}\Delta\mathcal{H}} + \mathcal{L}_t(\mathcal{W}_{t,t_{\text{te}}}^k) - \mathcal{L}_t^* + \mathcal{L}_{t_{\text{te}}}(\mathcal{W}_{t,t_{\text{te}}}^k) - \mathcal{L}_{t_{\text{te}}}^* \tag{31}$$

$$\leq d_{t,t_{\text{te}}}^{\mathcal{H}\Delta\mathcal{H}} + d_{t,t_{\text{te}}}^{\text{comb}} + \frac{1}{k} \tag{32}$$

$$= d_{t,t_{\text{te}}} + \frac{1}{k}. \tag{33}$$

Thus,

$$\left| \sum_{t=0}^{t_{\text{te}}-1} \alpha_t (\mathcal{L}_t(\mathcal{W}) - \mathcal{L}_t^*) - (\mathcal{L}_{t_{\text{te}}}(\mathcal{W}) - \mathcal{L}_{t_{\text{te}}}^*) \right| \tag{34}$$

$$= \left| \sum_{t=0}^{t_{\text{te}}-1} \alpha_t (\mathcal{L}_t(\mathcal{W}) - \mathcal{L}_t^*) - \sum_{t=0}^{t_{\text{te}}-1} \alpha_t (\mathcal{L}_{t_{\text{te}}}(\mathcal{W}) - \mathcal{L}_{t_{\text{te}}}^*) \right| \tag{35}$$

$$= \left| \sum_{t=0}^{t_{\text{te}}-1} \alpha_t ((\mathcal{L}_t(\mathcal{W}) - \mathcal{L}_t^*) - (\mathcal{L}_{t_{\text{te}}}(\mathcal{W}) - \mathcal{L}_{t_{\text{te}}}^*)) \right| \tag{36}$$

$$\leq \sum_{t=0}^{t_{\text{te}}-1} \alpha_t |(\mathcal{L}_t(\mathcal{W}) - \mathcal{L}_t^*) - (\mathcal{L}_{t_{\text{te}}}(\mathcal{W}) - \mathcal{L}_{t_{\text{te}}}^*)| \tag{37}$$

$$\leq \sum_{t=0}^{t_{\text{te}}-1} \alpha_t \left( d_{t,t_{\text{te}}} + \frac{1}{k} \right) \tag{38}$$

$$= \sum_{t=0}^{t_{\text{te}}-1} \alpha_t d_{t,t_{\text{te}}} + \frac{1}{k}. \tag{39}$$

Besides that, by Theorem 3 in [22] and Assumption 3,

$$\mathbb{P} \left\{ \sum_{t=0}^{t_{\text{te}}-1} \alpha_t \mathcal{L}^{\mathcal{D}_t}(\mathcal{W}) \geq \sum_{t=0}^{t_{\text{te}}-1} \alpha_t \mathcal{L}_t(\mathcal{W}) + \epsilon \right\} \tag{40}$$

$$\leq \exp \left( -\frac{\epsilon^2}{4 \sum_{t=0}^{t_{\text{te}}-1} m_t \left( \alpha_t \frac{\varsigma}{m_t} \right)^2} \right) = \exp \left( -\frac{\epsilon^2}{4 \varsigma^2 \sum_{t=0}^{t_{\text{te}}-1} \frac{\alpha_t^2}{m_t}} \right). \tag{41}$$

Then for $\varsigma = 1$, with probability at least $1 - \delta/2$,

$$\sum_{t=0}^{t_{\text{te}}-1} \alpha_t \mathcal{L}^{\mathcal{D}_t}(\mathcal{W}) \leq \sum_{t=0}^{t_{\text{te}}-1} \alpha_t \mathcal{L}_t(\mathcal{W}) + 2 \sqrt{\sum_{t=0}^{t_{\text{te}}-1} \frac{\alpha_t^2}{m_t} \log \frac{2}{\delta}}. \tag{42}$$

Similarly, with probability at least $1 - \delta/2$,

$$\sum_{t=0}^{t_{\text{te}}-1} \alpha_t \mathcal{L}_t(\mathcal{W}) \leq \sum_{t=0}^{t_{\text{te}}-1} \alpha_t \mathcal{L}^{\mathcal{D}_t}(\mathcal{W}) + 2 \sqrt{\sum_{t=0}^{t_{\text{te}}-1} \frac{\alpha_t^2}{m_t} \log \frac{2}{\delta}}. \tag{43}$$

Together, with probability at least $1 - \delta$,

$$\mathcal{L}_{t_{\text{te}}}(\mathcal{W}_{t_{\text{te}}-1}^{\boldsymbol{\alpha},\epsilon}) \tag{44}$$

$$= \mathcal{L}_{t_{\text{te}}}^* + \mathcal{L}_{t_{\text{te}}}(\mathcal{W}_{t_{\text{te}}-1}^{\boldsymbol{\alpha},\epsilon}) - \mathcal{L}_{t_{\text{te}}}^* \tag{45}$$

$$\leq \mathcal{L}_{t_{\text{te}}}^* + \sum_{t=0}^{t_{\text{te}}-1} \alpha_t (\mathcal{L}_t(\mathcal{W}_{t_{\text{te}}-1}^{\boldsymbol{\alpha},\epsilon}) - \mathcal{L}_t^*) + \sum_{t=0}^{t_{\text{te}}-1} \alpha_t d_{t,t_{\text{te}}} + \frac{1}{k} \tag{46}$$

$$\leq \mathcal{L}_{t_{\text{te}}}^* + \sum_{t=0}^{t_{\text{te}}-1} \alpha_t (\mathcal{L}^{\mathcal{D}_t}(\mathcal{W}_{t_{\text{te}}-1}^{\boldsymbol{\alpha},\epsilon}) - \mathcal{L}_t^*) + \sum_{t=0}^{t_{\text{te}}-1} \alpha_t d_{t,t_{\text{te}}}$$

$$+ \frac{1}{k} + 2 \sqrt{\sum_{t=0}^{t_{\text{te}}-1} \frac{\alpha_t^2}{m_t} \log \frac{2}{\delta}} \tag{47}$$

$$\leq \mathcal{L}_{t_{\text{te}}}^* + \epsilon + \sum_{t=0}^{t_{\text{te}}-1} \alpha_t (\mathcal{L}^{\mathcal{D}_t}(\mathcal{W}_{t_{\text{te}}}^k) - \mathcal{L}_t^*) + \sum_{t=0}^{T-1} \alpha_t d_{t,t_{\text{te}}}$$

$$+ \frac{1}{k} + 2 \sqrt{\sum_{t=0}^{t_{\text{te}}-1} \frac{\alpha_t^2}{m_t} \log \frac{2}{\delta}} \tag{48}$$

$$\leq \mathcal{L}_{t_{\text{te}}}^* + \epsilon + \sum_{t=0}^{t_{\text{te}}-1} \alpha_t (\mathcal{L}_t(\mathcal{W}_{t_{\text{te}}}^k) - \mathcal{L}_t^*) + \sum_{t=0}^{t_{\text{te}}-1} \alpha_t d_{t,t_{\text{te}}}$$

$$+ \frac{1}{k} + 4 \sqrt{\sum_{t=0}^{t_{\text{te}}-1} \frac{\alpha_t^2}{m_t} \log \frac{2}{\delta}} \tag{49}$$

$$\leq \mathcal{L}_{t_{\text{te}}}^* + \epsilon + \mathcal{L}_{t_{\text{te}}}(\mathcal{W}_{t_{\text{te}}}^k) - \mathcal{L}_t^* + 2 \sum_{t=0}^{t_{\text{te}}-1} \alpha_t d_{t,t_{\text{te}}} + \frac{2}{k} + 4 \sqrt{\sum_{t=0}^{t_{\text{te}}-1} \frac{\alpha_t^2}{m_t} \log \frac{2}{\delta}} \tag{50}$$

$$\leq \mathcal{L}_{t_{\text{te}}}^* + \epsilon + 2 \sum_{t=0}^{t_{\text{te}}-1} \alpha_t d_{t,t_{\text{te}}} + \frac{3}{k} + 4 \sqrt{\sum_{t=0}^{t_{\text{te}}-1} \frac{\alpha_t^2}{m_t} \log \frac{2}{\delta}}. \tag{51}$$

It follows from the continuity of probability that

$$\mathbb{P} \left\{ \mathcal{L}_{t_{\text{te}}}(\mathcal{W}_{t_{\text{te}}-1}^{\boldsymbol{\alpha},\epsilon}) > \mathcal{L}_{t_{\text{te}}}^* + \epsilon + 2 \sum_{t=0}^{t_{\text{te}}-1} \alpha_t d_{t,t_{\text{te}}} + 4 \sqrt{\sum_{t=0}^{t_{\text{te}}-1} \frac{\alpha_t^2}{m_t} \log \frac{2}{\delta}} \right\} \tag{52}$$

$$= \mathbb{P} \left[ \bigcup_{k=1}^{\infty} \left\{ \mathcal{L}_{t_{\text{te}}}(\mathcal{W}_{t_{\text{te}}-1}^{\boldsymbol{\alpha},\epsilon}) \geq \mathcal{L}_{t_{\text{te}}}^* + \epsilon + 2 \sum_{t=0}^{t_{\text{te}}-1} \alpha_t d_{t,t_{\text{te}}} \right. \right.$$

$$\left. \left. + \frac{3}{k} + 4 \sqrt{\sum_{t=0}^{t_{\text{te}}-1} \frac{\alpha_t^2}{m_t} \log \frac{2}{\delta}} \right\} \right] \tag{53}$$

$$= \lim_{k \to \infty} \mathbb{P} \left\{ \mathcal{L}_{t_{\text{te}}}(\mathcal{W}_{t_{\text{te}}-1}^{\boldsymbol{\alpha},\epsilon}) \geq \mathcal{L}_{t_{\text{te}}}^* + \epsilon + 2 \sum_{t=0}^{t_{\text{te}}-1} \alpha_t d_{t,t_{\text{te}}} \right.$$

$$\left. + \frac{3}{k} + 4 \sqrt{\sum_{t=0}^{t_{\text{te}}-1} \frac{\alpha_t^2}{m_t} \log \frac{2}{\delta}} \right\} \tag{54}$$

$$\leq \lim_{k \to \infty} \delta = \delta. \qquad \Box$$

**Corollary A.2.** *Under the setup of Lemma A.1, let*

$$L_{t_{\text{te}}}^{\boldsymbol{\alpha}} := \inf_{\substack{\epsilon > 0 \\ \epsilon \in \mathbb{Q}}} L_{t_{\text{te}}}(\mathcal{W}_{t_{\text{te}}-1}^{\boldsymbol{\alpha},\epsilon}) \tag{55}$$

denote the best possible loss w.r.t. $\boldsymbol{\alpha}$. With probability at least $1 - \delta$,

$$L_{t_{\text{te}}}^{\boldsymbol{\alpha}} \le \mathcal{L}_{t_{\text{te}}}^* + 2 \sum_{t=0}^{t_{\text{te}}-1} \alpha_t d_{t,t_{\text{te}}} + 4\sqrt{\sum_{t=0}^{t_{\text{te}}-1} \frac{\alpha_t^2}{m_t} \log \frac{2}{\delta}}. \tag{56}$$

Proof of Corollary A.2. By the continuity of probability,

$$\mathbb{P}\left\{ \mathcal{L}_{t_{\text{te}}}^{\boldsymbol{\alpha}} > \mathcal{L}_{t_{\text{te}}}^* + 2 \sum_{t=0}^{t_{\text{te}}-1} \alpha_t d_{tT} + 4\sqrt{\sum_{t=0}^{t_{\text{te}}-1} \frac{\alpha_t^2}{m_t} \log \frac{2}{\delta}} \right\} \tag{57}$$

$$= \mathbb{P}\left[ \bigcup_{k=1}^{\infty} \left\{ \mathcal{L}_{t_{\text{te}}}^{\boldsymbol{\alpha}} \ge \mathcal{L}_{t_{\text{te}}}^* + \frac{1}{k} + 2 \sum_{t=0}^{t_{\text{te}}-1} \alpha_t d_{t,t_{\text{te}}} + 4\sqrt{\sum_{t=0}^{t_{\text{te}}-1} \frac{\alpha_t^2}{m_t} \log \frac{2}{\delta}} \right\} \right] \tag{58}$$

$$= \lim_{k \to \infty} \mathbb{P}\left\{ \mathcal{L}_{t_{\text{te}}}^{\boldsymbol{\alpha}} \ge \mathcal{L}_{t_{\text{te}}}^* + \frac{1}{k} + 2 \sum_{t=0}^{t_{\text{te}}-1} \alpha_t d_{t,t_{\text{te}}} + 4\sqrt{\sum_{t=0}^{t_{\text{te}}-1} \frac{\alpha_t^2}{m_t} \log \frac{2}{\delta}} \right\} \tag{59}$$

$$\le \limsup_{k \to \infty} \mathbb{P}\left\{ \mathcal{L}_{t_{\text{te}}}\left(\mathcal{W}_{t_{\text{te}}-1}^{\boldsymbol{\alpha}, \frac{1}{k}}\right) \ge \mathcal{L}_{t_{\text{te}}}^* + \frac{1}{k} + 2 \sum_{t=0}^{t_{\text{te}}-1} \alpha_t d_{t,t_{\text{te}}} \right.$$

$$\left. + 4\sqrt{\sum_{t=0}^{t_{\text{te}}-1} \frac{\alpha_t^2}{m_t} \log \frac{2}{\delta}} \right\} \tag{60}$$

$$\le \limsup_{k \to \infty} \delta = \delta. \qquad \square$$

Now we give the proofs of Theorems 3.1 & 3.2.

Proof of Theorem 3.1. By Assumption 4,

$$\ell_{t_{\text{te}}-1}(\mathcal{W}) = \mathcal{L}^{\mathcal{D}_{t_{\text{te}}-1}}(\mathcal{W}) + \gamma \ell_{t_{\text{te}}-2}(\mathcal{W}) = \sum_{t=0}^{t_{\text{te}}-1} \gamma^{t_{\text{te}}-t-1} \mathcal{L}^{\mathcal{D}_t}(\mathcal{W}). \tag{61}$$

Thus, normalizing the coefficients gives

$$\alpha_t^{\text{ft}} := \frac{(1-\gamma)\gamma^{t_{\text{te}}-t-1}}{1 - \gamma^{t_{\text{te}}}}. \tag{62}$$

It follows from Corollary A.2 that

$$\mathcal{L}_{t_{\text{te}}}^{\text{ft}} = \mathcal{L}_{t_{\text{te}}}^{\boldsymbol{\alpha}^{\text{ft}}} \tag{63}$$

$$\le \mathcal{L}_{t_{\text{te}}}^* + 2 \sum_{t=0}^{t_{\text{te}}-1} \alpha_t^{\text{ft}} d_{t,t_{\text{te}}} + 4\sqrt{\sum_{t=0}^{t_{\text{te}}-1} \frac{(\alpha_t^{\text{ft}})^2}{m_t} \log \frac{2}{\delta}} \tag{64}$$

$$= \mathcal{L}_{t_{\text{te}}}^* + \frac{(1-\gamma)\left(2\sum_{t=0}^{t_{\text{te}}-1} \gamma^{t_{\text{te}}-t-1} d_{t,t_{\text{te}}} + 4\sqrt{\left(\frac{\gamma^{2t_{\text{te}}-2}}{m_0} + \frac{1-\gamma^{2t_{\text{te}}-2}}{(1-\gamma^2)m_1}\right)\log \frac{2}{\delta}}\right)}{1 - \gamma^{t_{\text{te}}}} \quad \square$$

Proof of Theorem 3.2. By Assumption 5, we have

$$\alpha_t^{\text{rt}} := \frac{m_t}{\sum_{t'=0}^{t_{\text{te}}-1} m_{t'}} = \frac{m_t}{m_0 + (t_{\text{te}} - 1)m_1}. \tag{65}$$

It follows from Corollary A.2 that

$$\mathcal{L}_{t_{\text{te}}}^{\text{rt}} = \mathcal{L}_{t_{\text{te}}}^{\boldsymbol{\alpha}^{\text{rt}}} \tag{66}$$

$$\le \mathcal{L}_{t_{\text{te}}}^* + 2 \sum_{t=0}^{t_{\text{te}}-1} \alpha_t^{\text{rt}} d_{t,t_{\text{te}}} + 4\sqrt{\sum_{t=0}^{t_{\text{te}}-1} \frac{(\alpha_t^{\text{rt}})^2}{m_t} \log \frac{2}{\delta}} \tag{67}$$

$$= \mathcal{L}_{t_{\text{te}}}^* + \frac{2m_0 d_{0T} + 2\sum_{t=1}^{t_{\text{te}}-1} m_1 d_{t,t_{\text{te}}}}{m_0 + (t_{\text{te}} - 1)m_1} + 4\sqrt{\frac{1}{m_0 + (t_{\text{te}} - 1)m_1} \log \frac{2}{\delta}}. \quad \square$$

## A.3 Proof of Proposition 3.3

Proof of Proposition 3.3. Note that

$$\nabla_{\mathcal{W}_t} \mathcal{L}_{\text{fair}}^{\mathcal{D}_t}(\mathcal{W}_t) = \nabla_{\mathcal{W}_t}(-\log(\sigma(-\text{DPD}^{\mathcal{D}_t}(\mathcal{W}_t)))) \tag{68}$$

$$= \sigma(\text{DPD}^{\mathcal{D}_t}(\mathcal{W}_t))\nabla_{\mathcal{W}_t} \text{DPD}^{\mathcal{D}_t}(\mathcal{W}_t). \tag{69}$$

Since $\nabla_{\mathcal{W}_t} \mathcal{L}_{\text{fair}}^{\mathcal{D}_t}(\mathcal{W}_t) \ne \mathbf{0}$, then $\nabla_{\mathcal{W}_t} \text{DPD}^{\mathcal{D}_t}(\mathcal{W}_t) \ne \mathbf{0}$. Consider

$$\lambda := \frac{-2\langle \nabla_{\mathcal{W}_t} \mathcal{L}_{\text{rec}}^{\mathcal{D}_t}(\mathcal{W}_t), \nabla_{\mathcal{W}_t} \text{DPD}^{\mathcal{D}_t}(\mathcal{W}_t)\rangle}{\|\nabla_{\mathcal{W}_t} \text{DPD}^{\mathcal{D}_t}(\mathcal{W}_t)\|_2^2} \ge 0. \tag{70}$$

By the chain rule,

$$\lim_{\eta \to +0} \frac{\text{DPD}^{\mathcal{D}_t}(\widetilde{\mathcal{W}_t}) - \text{DPD}^{\mathcal{D}_t}(\mathcal{W}_t)}{\eta} \tag{71}$$

$$= \langle \nabla_{\mathcal{W}_t} \mathcal{L}_{\text{rec}}^{\mathcal{D}_t}(\mathcal{W}_t) + \lambda \nabla_{\mathcal{W}_t} \mathcal{L}_{\text{fair}}^{\mathcal{D}_t}(\mathcal{W}_t), \nabla_{\mathcal{W}_t} \text{DPD}^{\mathcal{D}_t}(\mathcal{W}_t)\rangle \tag{72}$$

$$= (1 - 2\sigma(\text{DPD}^{\mathcal{D}_t}(\mathcal{W}_t)))\langle \nabla_{\mathcal{W}_t} \mathcal{L}_{\text{rec}}^{\mathcal{D}_t}(\mathcal{W}_t), \nabla_{\mathcal{W}_t} \text{DPD}^{\mathcal{D}_t}(\mathcal{W}_t)\rangle. \tag{73}$$

The conclusion follows from the fact that

$$\text{sgn}(x)(1 - 2\sigma(x)) \le 0, \qquad \forall x \in \mathbb{R}. \tag{74}$$

$$\square$$

## B EXPERIMENTS

## B.1 Implementation Details of Competitors

For Adver, the adversarial coefficient $\gamma$ is selected from the suggested range $[1, 10, 20, 50]$, as mentioned in their paper [18]. The filter modules are two-layer neural networks with the LeakyReLU activation. The discriminators are multi-layer perceptrons with 7 layers, LeakyReLU activation function, and a dropout rate of 0.3. The discriminators are trained for 10 steps.

In the original paper of Rerank [17], they use a re-ranking technique under a fairness-constraint based on the test positive data, which does not align with our assumption that we cannot access future data when serving the recommendation list. Thus, we adopt this method by designating items with predicted scores above a certain threshold as ground-truth items. In our experiments, the predicted scores are normalized to the range of 0 to 1, and we set the threshold to 0.7.

## B.2 Software and Hardware Configuration.

All codes are programmed in Python 3.6.9 and PyTorch 1.4.0. All experiments are performed on a Linux server with 2 Intel Xeon Gold 6240R CPUs and 1 Nvidia Tesla V100 SXM2 GPU with 32 GB GPU memory.

## B.3 Additional Effectiveness Results

- Fig. 6 show the results for the trade-off between recommendation performance and absolute performance disparity in Task-N. The results for Task-R is in the main body.
- Fig. 7 shows the results for the trend of performance disparity in Task-N. The results for Task-R is in the main body.
- Fig. 9 and Fig. 8 show the trend of recommendation performance in Task-R and Task-N, respectively.
- Fig. 10 displays the trend in recommendation performance of PRETRAIN, RETRAIN, FINETUNE, RETRAIN-FAIR, and FADE in Task-R on Movielenz. This includes the results immediately after pretraining (i.e., $t = 0$) and subsequent time periods (i.e., $t = 7, 8, 9$). Notably, the results for MF demonstrate that fine-tuning-based methods outperform retraining-based methods in the earlier time periods because fine-tuning is less affected by distribution shifts. However, in later periods, the performance of fine-tuned models eventually degrades, falling even below that of retrained models due to accumulated learning errors. These observations are consistent with our theoretical analyses in §3.1 and suggest that the best practice involves incremental fine-tuning with restart.

## B.4 Comparison of Soft Ranking Methods

Fig. 11 presents FADE adapting different soft ranking metrics, including ApproxNDCG [26] and NeuralNDCG [25], as well as FADE incorporating the differentiable Hit in Task-R. The legend also provides the average running time for each method.

First, FADE outperforms or matches the NeuralNDCG variant in both recommendation performance and performance disparity, while being approximately four times faster. This is because the differentiable Hit addresses NeuralNDCG's gradient vanishing issue by eliminating several processes, including the sinkhorn algorithm.

In comparison to ApproxNDCG, FADE generally achieves smaller performance disparity. Although ApproxNDCG may yield lower disparity in some cases, it excessively sacrifices recommendation quality, which is undesirable.

## B.5 Hyperparameter Analysis

*B.5.1 Effect of the scaling parameter $\lambda$ for the fairness Loss.* Fig. 13 and Fig. 17 show the effect of the scaling parameter $\lambda$ on the recommendation performances of the advantaged and disadvantaged groups in Task-R and Task-N, respectively.

*B.5.2 Effect of the number of dynamic update epochs.* Fig. 14 and Fig. 18 show the effect of the number of dynamic update epochs on the recommendation performances of the advantaged and disadvantaged groups in Task-R and Task-N, respectively.

*B.5.3 Effect of temperature parameter $\tau$ in the relaxed permutation matrix.* Fig. 15 and Fig. 19 show the effect of the temperature parameter $\tau$ on the recommendation performances of the advantaged and disadvantaged groups in Task-R and Task-N, respectively.

*B.5.4 Effect of the number of negative items $\mu$.* Fig. 16 and Fig. 20 show the effect of the number of negative candidate items $\mu$ for a user in our fairness loss on the recommendation performances of the advantaged and disadvantaged groups in Task-R and Task-N, respectively.

In general, the results suggest that FADE performance remains relatively stable when varying the number of negative items in most cases. Thus, setting $\mu$ to 4 results in comparable performance while also enhancing the execution time of FADE.

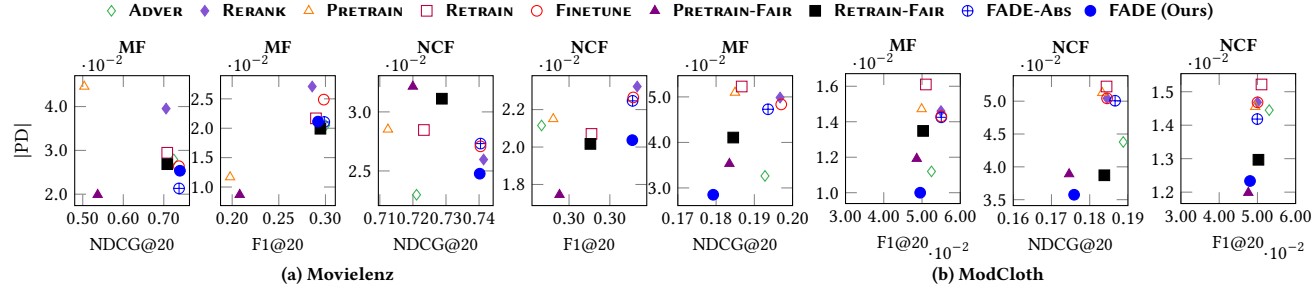

**Figure 6: Trade-off between recommendation performance and absolute performance disparity in Task-N.**

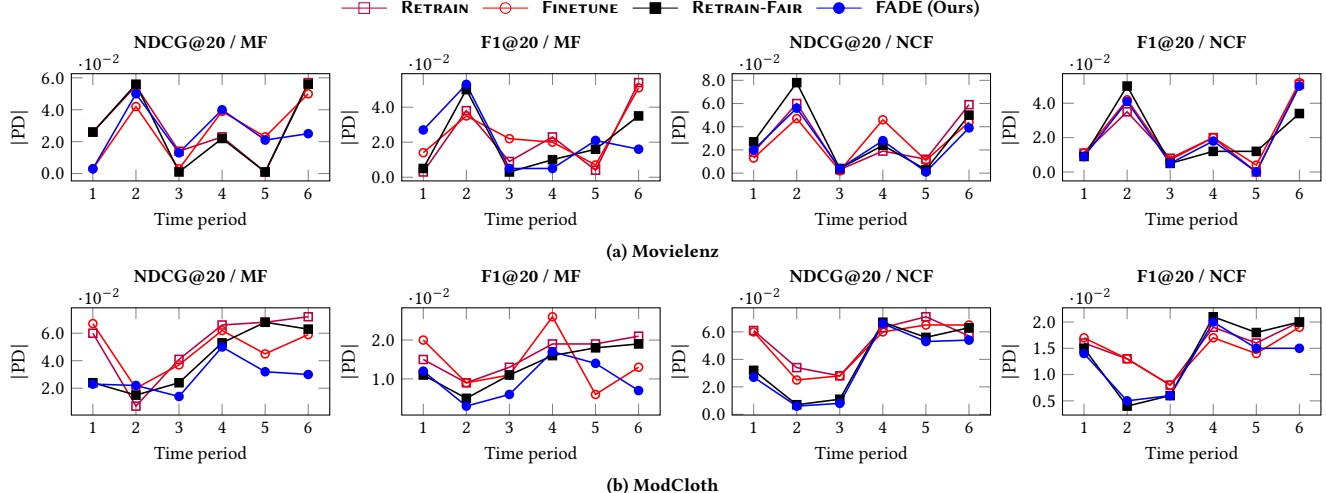

**Figure 7: Trend of absolute performance disparity in Task-N.**

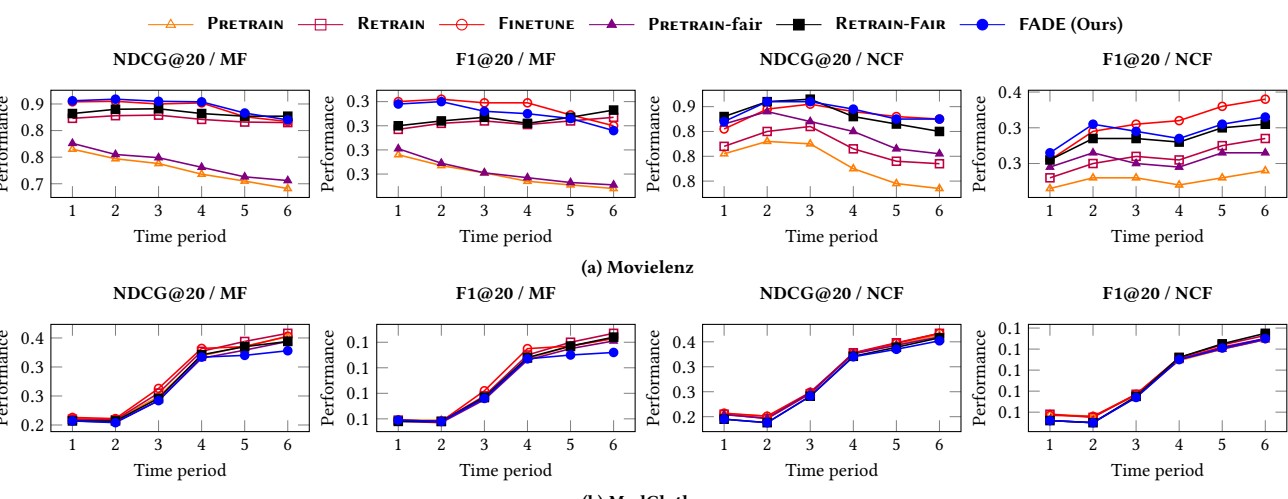

**Figure 8: Trend of recommendation performance in Task-R.**

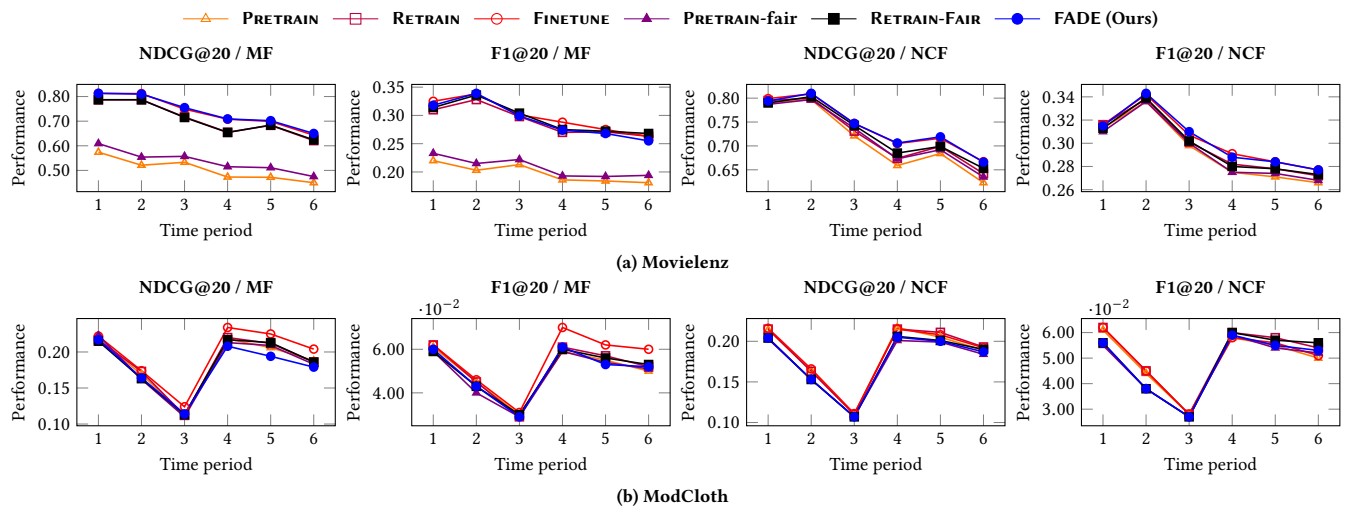

Figure 9: Trend of recommendation performance in Task-N.

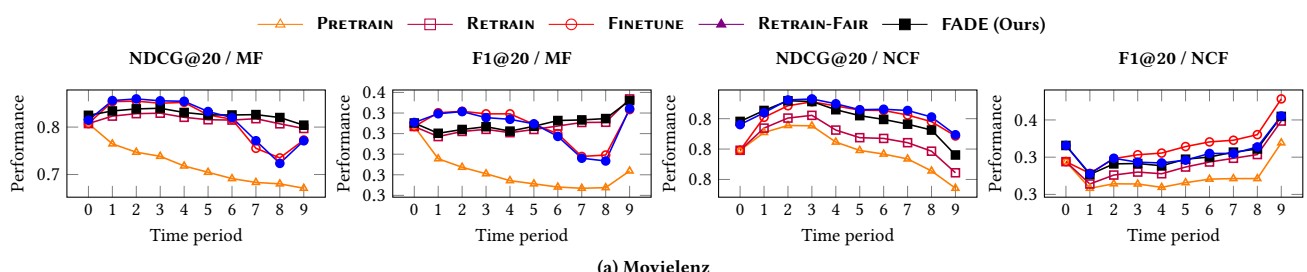

Figure 10: Trend of recommendation performance in Task-R on Movielenz, including subsequent time periods.

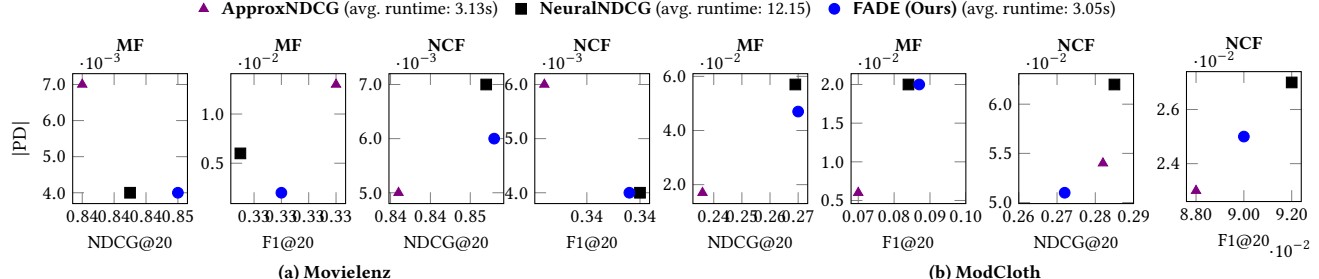

Figure 11: Trade-off between recommendation performance and absolute performance disparity in Task-R.

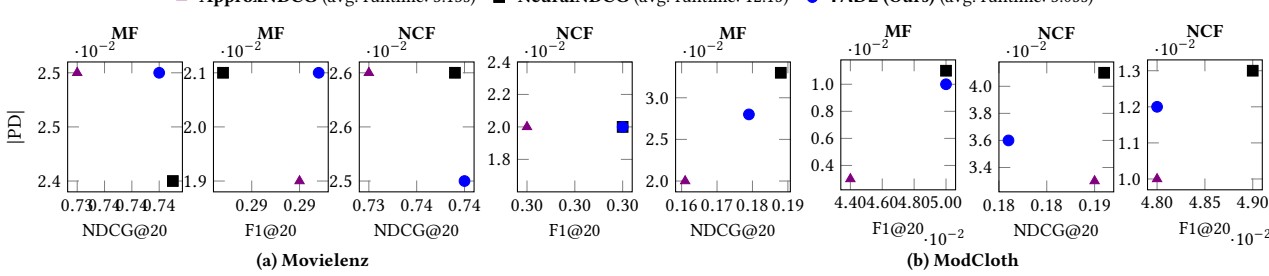

Figure 12: Trade-off between recommendation performance and absolute performance disparity in Task-N.

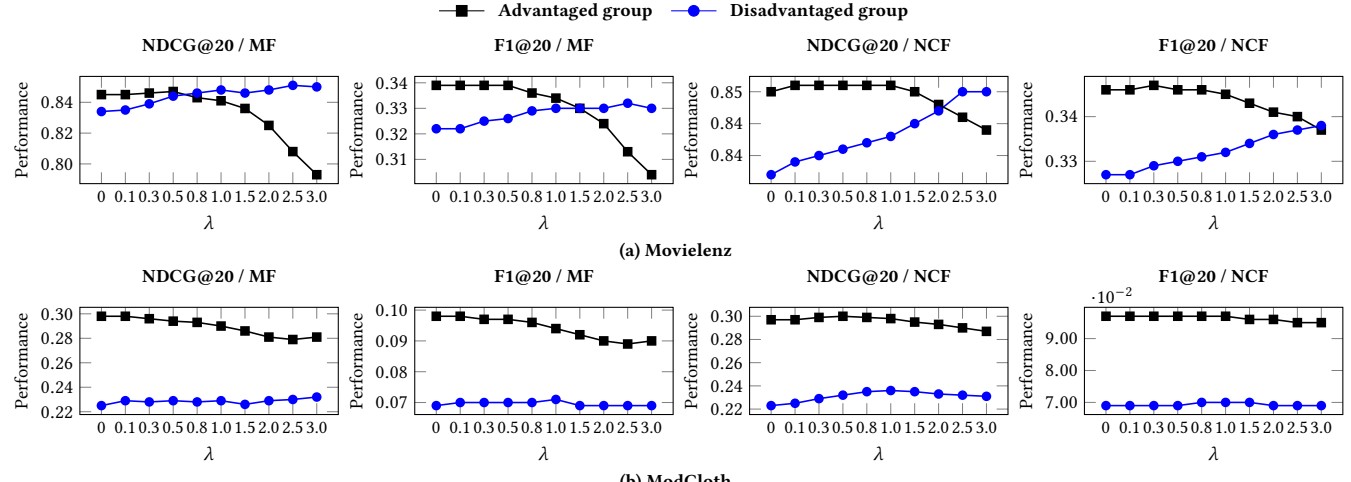

Figure 13: The effect of the scaling factor $\lambda$ on the recommendation performances of the advantaged and disadvantaged groups in Task-R.

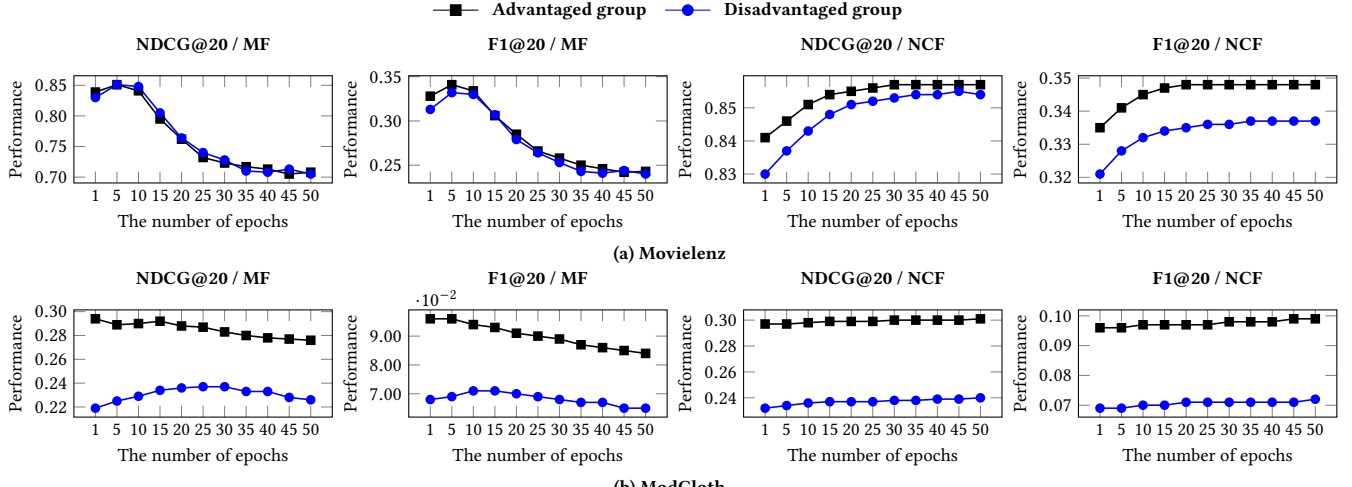

Figure 14: The effect of the number of dynamic update epochs on the recommendation performances of the advantaged and disadvantaged groups in Task-R.

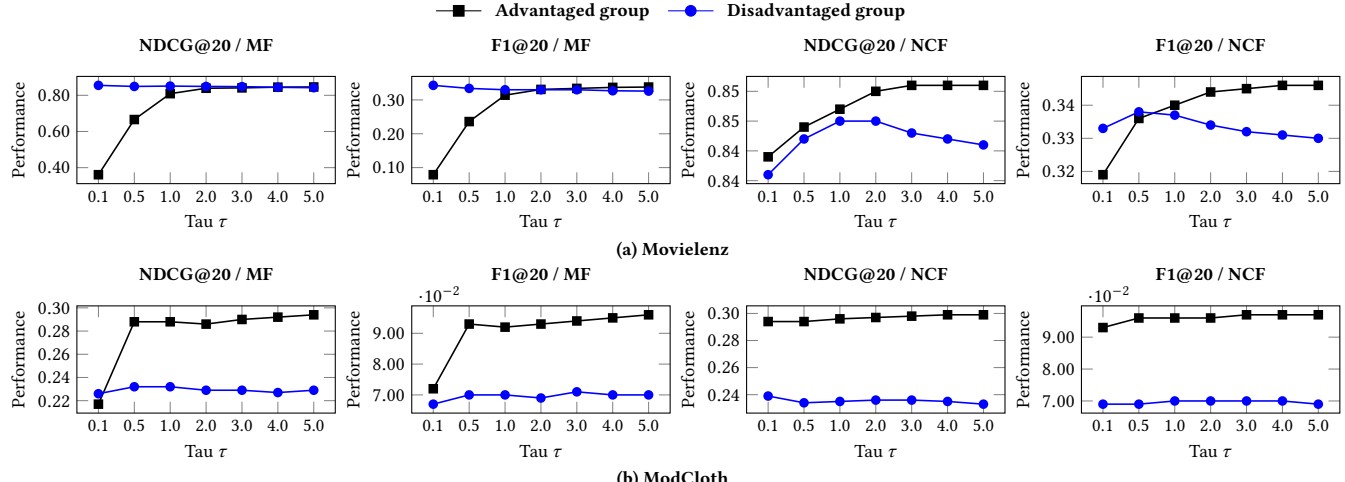

Figure 15: The effect of the hyperparameter $\tau$ in our Differentiable Hit (DH) on the recommendation performances of the advantaged and disadvantaged groups in Task-R.

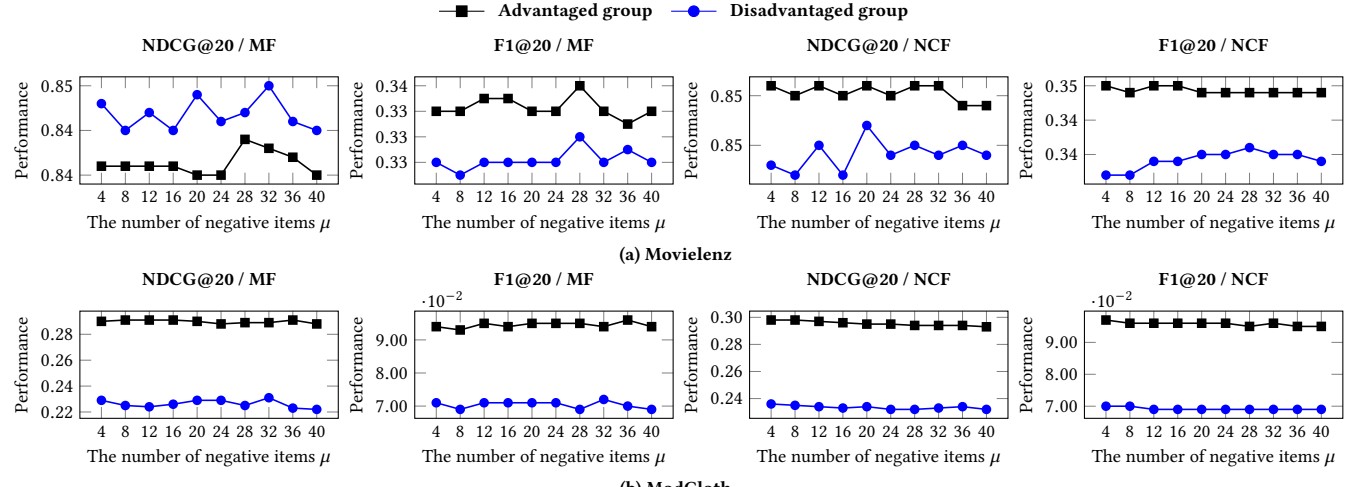

Figure 16: The effect of the number of negative items for each user in our fairness loss on the recommendation performances of the advantaged and disadvantaged groups in Task-R.

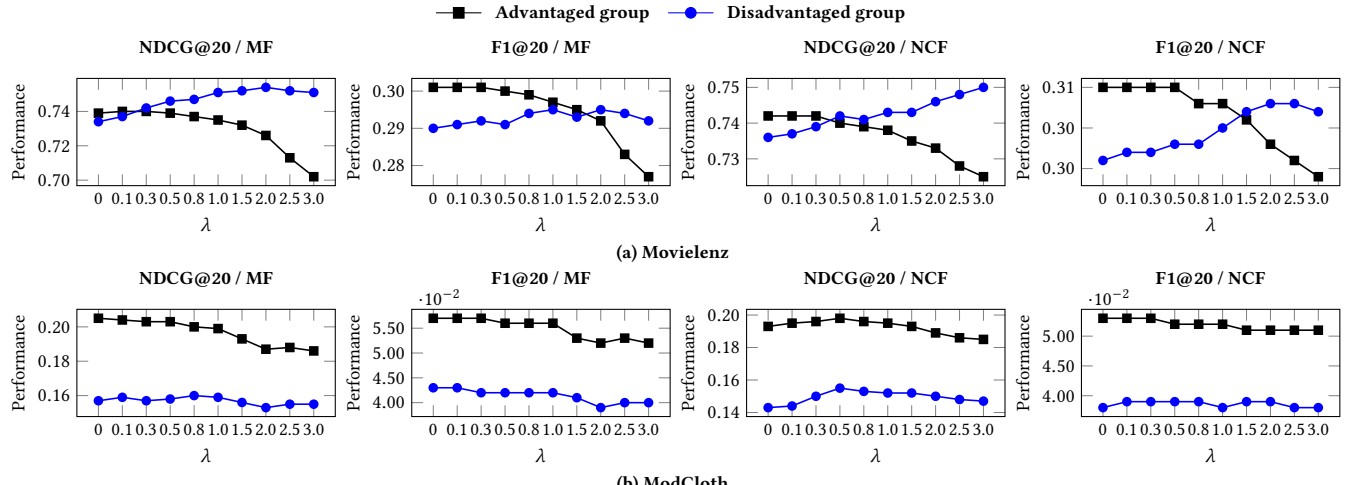

Figure 17: The effect of the scaling factor $\lambda$ on the recommendation performances of the advantaged and disadvantaged groups in Task-N.

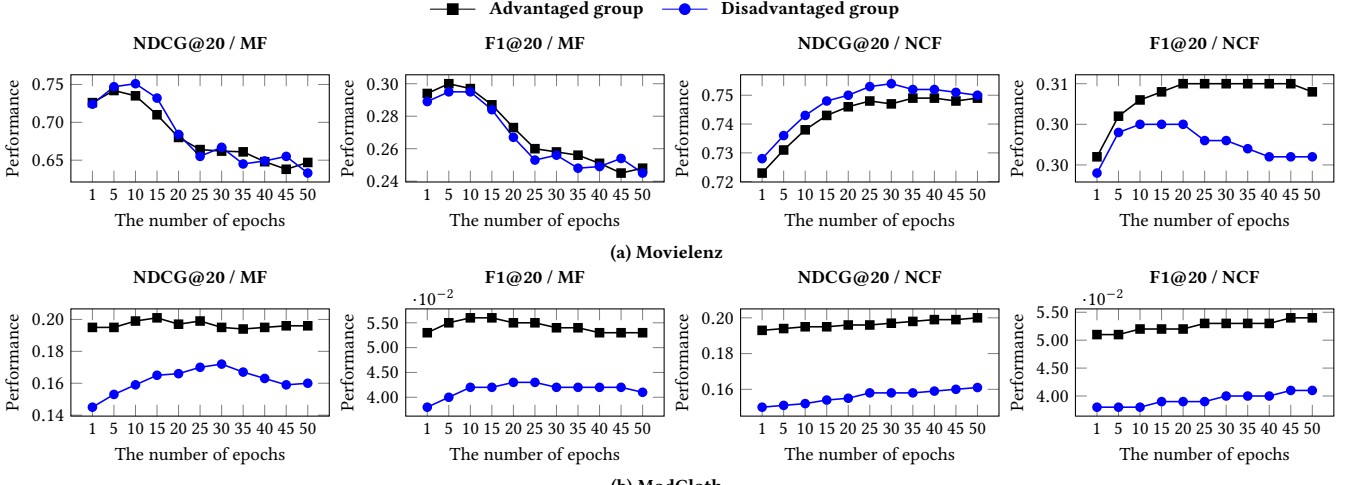

Figure 18: The effect of the number of dynamic update epochs on the recommendation performances of the advantaged and disadvantaged groups in Task-N.

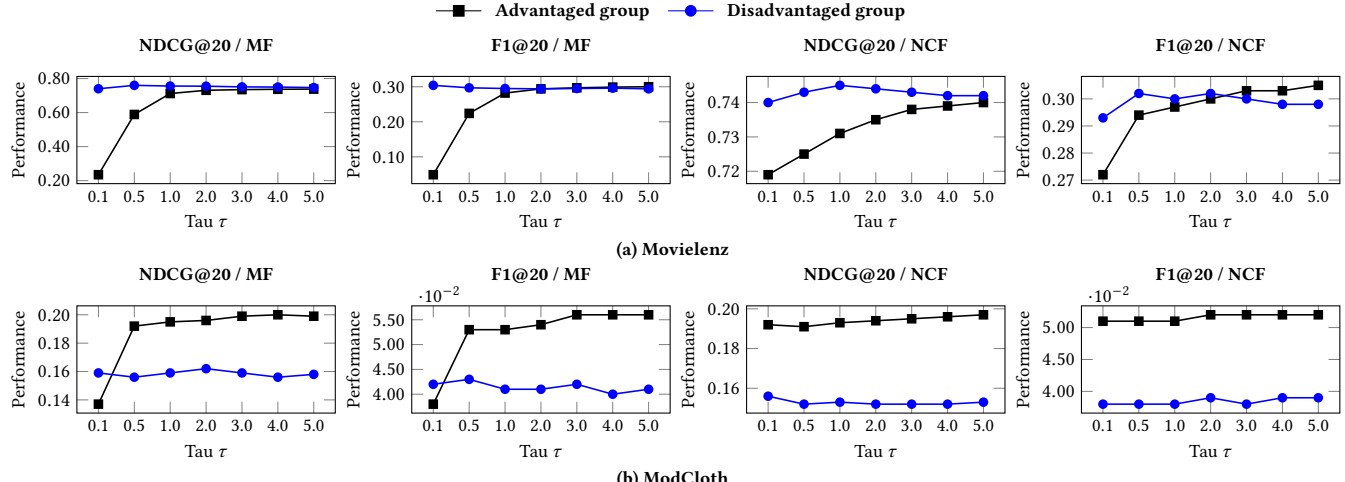

Figure 19: The effect of the hyperparameter $\tau$ in our Differentiable Hit (DH) on the recommendation performances of the advantaged and disadvantaged groups in Task-N.

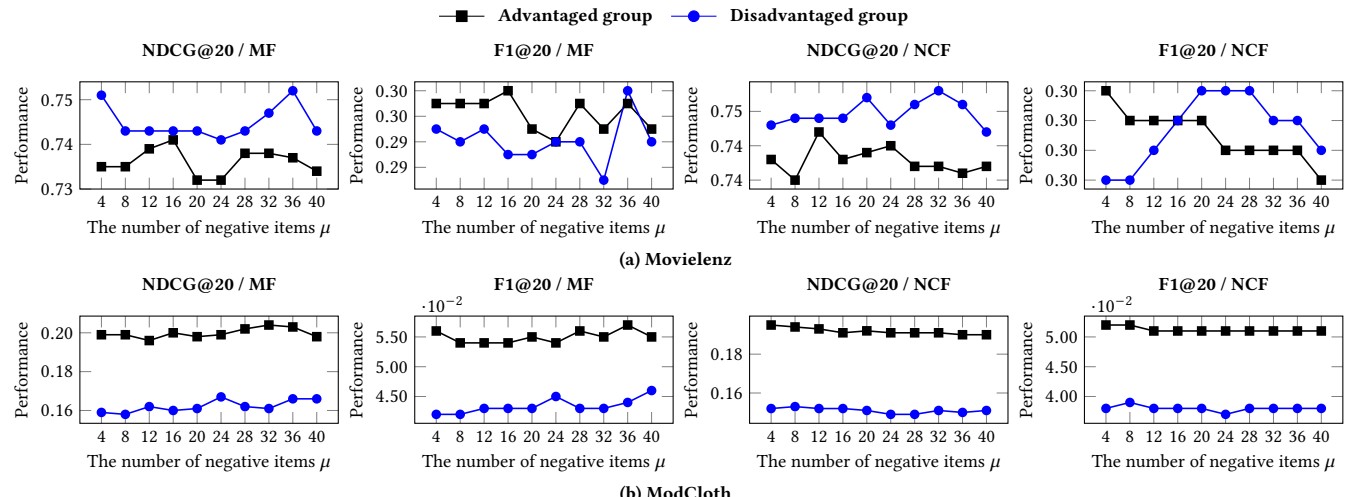

Figure 20: The effect of the number of negative items for each user in our fairness loss on the recommendation performances of the advantaged and disadvantaged groups in Task-N.

