# OpenReview forum: "Ensuring User-side Fairness in Dynamic Recommender Systems"
_ACM.org/TheWebConf/2024/Conference — TheWebConf24 Oral_

### Official Review · Reviewer_uFf9 · 2023-11-10

**Novelty:** 3
**Technical Quality:** 3

**Review:**

In summary, while the author introduces an intriguing topic addressing user fairness under distribution shift, upon further examination, I find that the paper fails in effectively addressing the issue, particularly with the proposed modules such as BPR-based loss and the challenge posed by the non-differentiability of ranking metrics. The fine-tune technique seems to be irrelevant to the user fairness problem. In simpler terms, I find myself perplexed as to whether C1-C3 effectively and coherently address user fairness. I hope the author can answer the questions to address my concerns.

**Questions:**

(1)	It appears that the initial work may be making an overclaim in its attempt to address dynamic user fairness. See [1] focused on the time-vary recommendation attributes of items/users. [2] focused on feature shifts for fairness. [3] also proposed a debias method to focus on changing distribution for the testing and training phase and softened the ranking metrics as a fully differentiable framework. While some of the existing research may not explicitly focus on user fairness, it is imperative to recognize that the differential privacy (DP) metric and feature shift, though seemingly distinct, share similarities and warrant a comparative analysis in evaluating the overall fairness implications of a model.
(2)	I carefully examine the Proof of Theorem 3.1 and 3.2, however, I cannot find the relationship between fairness and the theorems. If I miss the correlation, please point out which lines in the proof support the correlation
(3)	If the problem (2) has a correlation, how is the method of BPR-related loss and differentiable hit well to solve the unfairness of the fine-tuning phase? It seems they are an independent part. How does your method well address dynamic user-fairness uniquely?
(4)	As you mentioned, baselines Adver's primary focus is not on reducing the performance disparity among different user groups. More baselines for addressing feature shift should be added such as the IPS[3] method or some user-fairness-aware methods.
(5)	Why change the metric from DPD to PD in the experiment?
(6)	In lines 891-892, there is also some work to regard the DP of utilities metric as item fairness[4]. It is better to further address the fairness concept.
[1] Ge, Y., Liu, S., Gao, R., Xian, Y., Li, Y., Zhao, X., ... & Zhang, Y. (2021, March). Towards long-term fairness in the recommendation. In Proceedings of the 14th ACM International Conference on web search and data mining (pp. 445-453).
[2] Nil-Jana Akpinar, Cyrus DiCiccio, Preetam Nandy, and Kinjal Basu. 2022. Long-term Dynamics of Fairness Intervention in Connection Recommender Systems. In Proceedings of the 2022 AAAI/ACM Conference on AI, Ethics, and Society (AIES '22). Association for Computing Machinery, New York, NY, USA, 22–35. https://doi.org/10.1145/3514094.3534173
[3] Jiakai Tang, Shiqi Shen, Zhipeng Wang, Zhi Gong, Jingsen Zhang, and Xu Chen. 2023. When Fairness meets Bias: a Debiased Framework for Fairness aware Top-N Recommendation. In Proceedings of the 17th ACM Conference on Recommender Systems (RecSys '23). Association for Computing Machinery, New York, NY, USA, 200–210. https://doi.org/10.1145/3604915.3608770
[4] Wu, Y., Cao, J., Xu, G., & Tan, Y. (2021, July). Tfrom: A two-sided fairness-aware recommendation model for both customers and providers. In Proceedings of the 44th International ACM SIGIR Conference on Research and Development in Information Retrieval (pp. 1013-1022).

**Reviewer Confidence:**

4: The reviewer is certain that the evaluation is correct and very familiar with the relevant literature

**Scope:**

3: The work is somewhat relevant to the Web and to the track, and is of narrow interest to a sub-community

---

### Official Review · Reviewer_Sa9v · 2023-11-23

**Novelty:** 4
**Technical Quality:** 5

**Review:**

This paper investigates the challenge of user-side fairness in dynamic recommendation scenarios, proposing the FADE framework to dynamically ensure fairness over time. FADE incorporates a fairness loss equipped with the lightweight Differentiable Hit, addressing the gradient vanishing issue present in the recent NeuralNDCG method and enhancing overall efficiency.

Strengths:

S1: This is the inaugural research endeavor addressing the critical matter of ensuring user-side fairness in dynamic recommendation scenarios.

S2: The motivation behind this work is validated through a thorough theoretical analysis, comparing fine-tuning and retraining in terms of generalization error (recommendation and fairness) under distribution shift.

S3: Technically, the incremental fine-tuning is introduced effectively balancing both fairness and utility. The differentiable hit is conducted to overcome the non-differentiability of recommendation metrics in the fairness loss.

S4: Extensive experiments conducted on real-world datasets demonstrate that FADE successfully and efficiently mitigates performance disparities with minimal sacrifice to overall recommendation performance.

Weaknesses:

The paper lacks some related references.

[1] Chen X, Fan W, Chen J, et al. Fairly adaptive negative sampling for recommendations[C]//Proceedings of the ACM Web Conference 2023. 2023: 3723-3733.
[2] Dong Y, Kang J, Tong H, et al. Individual fairness for graph neural networks: A ranking based approach[C]//Proceedings of the 27th ACM SIGKDD Conference on Knowledge Discovery & Data Mining. 2021: 300-310.

**Questions:**

I do not have specific questions. However, I am not convinced by the reproducibility or resource availability of this work, so if the authors can provide more guidance on that, that would be appreciated.

**Ethics Review Description:**

No ethical issues.

**Reviewer Confidence:**

2: The reviewer is willing to defend the evaluation, but it is likely that the reviewer did not understand parts of the paper

**Scope:**

3: The work is somewhat relevant to the Web and to the track, and is of narrow interest to a sub-community

---

### Official Review · Reviewer_n3X7 · 2023-11-24

**Novelty:** 6
**Technical Quality:** 6

**Review:**

In this submission, the authors propose FAir Dynamic rEcommender (FADE), an end-to-end fine-tuning framework to dynamically ensuring user-side fairness over time.

Pros: 1) The paper is well-written and easy to read; 2) The research problem is important and interesting; 3) The experiment section is comprehensive.

Cons: 1) Only F1/NDCG@20 results are reported, more results with different lengths should be included. Also, only one fairness metric has been reported; 2) Clarification is needed regarding whether the baselines also periodically update their model parameters. If they do, it would be helpful to see these results in a graphical format. If not, the basis for comparison may not be equitable. 3) Can you explain the reason that fade-abs performs much worse than fade?

**Questions:**

Line 361, “We will define Lfair in §4.3.” should be “We will define Lfair in §3.4.”

**Reviewer Confidence:**

3: The reviewer is confident but not certain that the evaluation is correct

**Scope:**

3: The work is somewhat relevant to the Web and to the track, and is of narrow interest to a sub-community

---

### Official Review · Reviewer_S3eQ · 2023-11-24

**Novelty:** 4
**Technical Quality:** 5

**Review:**

## Summary
This paper is concerned with a fairness-aware method for dynamic recommendation.
I believe the paper would significantly benefit from further developing the discussion in Section 3.1 to align more closely with the setting of recommendation systems. Sections 3.3 and 3.4 seem to be completely unrelated to this discussion and offer little novelty. Therefore, I strongly recommend revising the paper to reflect this.

## Strengths
(S1) The paper is well-written and dense.


(S2) The authors make considerable effort to present the proposed method using various illustrations.


(S3) The authors provide theoretical analyses.


## Weakness
(W1) While I found the paper to be technically solid, it appears to employ only existing methods simultaneously to address the issue of user-side fairness in dynamic recommendation. For instance, the use of differentiable ranking measures doesn't seem to deviate from existing settings in either the context of fairness or dynamic recommendation and the fairness measures also appear conventional. It would be interesting if these metrics under an incremental fine-tuning strategy differ from other settings, but as it stands, the paper seems to merely combine them without a clear necessity for this specific approach.


(W2) It is commendable that the authors provide a generalization error analysis in a complex setting. However, my understanding is that the paper deals with a dynamic but not "sequential" context. If Assumption 1 assumes that data at each time point are i.i.d., this seems to merely repeat learning under distribution shift multiple times. Without a weaker assumption that preserves the sequential nature, I cannot see how this setting introduces unique challenges compared to one-time learning under existing distribution shifts. However, I did find the discussion on retraining after certain periods based on the obtained results quite interesting (though the conclusion itself seems trivial without assumptions on the non-i.i.d. nature and the magnitude of distribution shifts at each time point). Also, I'm not quite clear why the authors evaluate only the error bound up to $t_{te}-1$ instead of the regret bound. Practically, one often wants to minimize the regret bound. In that case, is it possible to derive optimal $T$ (i.e., the interval of retraining) for minimizing regret?

(W3) The analysis in Section 3.1 seems more applicable to a general machine-learning setting than the specific setting of this paper. Therefore, if the paper claims novelty or nontriviality of this theory, I believe it should be judged in an ML-focused conference (e.g., NeurIPS, ICML, ICLR).

**Questions:**

## Questions
(Q1) Is there a comment about the above weaknesses from the authors' perspective?

**Reviewer Confidence:**

2: The reviewer is willing to defend the evaluation, but it is likely that the reviewer did not understand parts of the paper

**Scope:**

3: The work is somewhat relevant to the Web and to the track, and is of narrow interest to a sub-community

---

### Decision · Program_Chairs · 2024-01-22

**Decision:**

Accept (Oral)

**Comment:**

The paper aims at the user-side fairness problem of recommender systems under the dynamic setting, which aims to guarantee fairness even if the model and environments are updated. To achieve this goal, the paper focus on three main challenges: distribution shifts, frequent model updates, and non-differentiability of ranking metrics. To solve the problems, the paper contributes several novel innovations such as differentiable hit, fast and effective soft ranking metric, and fairness loss without absolute. The paper received comprehensive discussions among reviewers and authors. During discussions, authors made further clarifications of the technical contribution and difference from existing work, and also reported results on more baseline methods suggested by reviewers. Overall, the paper tackles a new and important problem, proposed novelty methods involving differentiable ranking and incremental learning, and provided both theoretical and empirical evidences for the proposed method. Authors are encouraged to integrate the discussions and clarifications into the paper.